# MedJourney: Benchmark and Evaluation of Large Language Models over Patient Clinical Journey

**Xian Wu[1], Yutian Zhao[1], Yunyan Zhang[1], Jiageng Wu[2], Zhihong Zhu[3]**
**Yingying Zhang[1], Yi Ouyang[1], Ziheng Zhang[1], Huimin Wang[1], Zhenxi Lin[1]**
**Jie Yang[4], Shuang Zhao[5], Yefeng Zheng[6]** [*]

[1]Tencent Youtu Lab, Jarvis Research Center    [2]Zhejiang University
[3]Peking University   [4]Harvard Medical School
[5] Xiangya Hospital   [6] Westlake University
{kevinxwu, yutianzhao, yunyanzhang}@tencent.com
jiagengwu@zju.edu.cn, zhihongzhu@stu.pku.edu.cn
{ninzhang, yiouyang, zihengzhang, chalerislin}@tencent.com
jieynlp@gmail.com, shuangxy@csu.edu.cn, yefengzheng@westlake.com.cn

## Abstract

Large language models (LLMs) have demonstrated remarkable capabilities in language understanding and generation, leading to their widespread adoption across various fields. Among these, the medical field is particularly well-suited for LLM applications, as many medical tasks can be enhanced by LLMs. Despite the existence of benchmarks for evaluating LLMs in medical question-answering and exams, there remains a notable gap in assessing LLMs' performance in supporting patients throughout their entire hospital visit journey in real-world clinical practice. In this paper, we address this gap by dividing a typical patient's clinical journey into four stages: planning, access, delivery and ongoing care. For each stage, we introduce multiple tasks and corresponding datasets, resulting in a comprehensive benchmark comprising 12 datasets, of which five are newly introduced, and seven are constructed from existing datasets. This proposed benchmark facilitates a thorough evaluation of LLMs' effectiveness across the entire patient journey, providing insights into their practical application in clinical settings. Additionally, we evaluate three categories of LLMs against this benchmark: 1) proprietary LLM services such as GPT-4; 2) public LLMs like QWen; and 3) specialized medical LLMs, like HuatuoGPT2. Through this extensive evaluation, we aim to provide a better understanding of LLMs' performance in the medical domain, ultimately contributing to their more effective deployment in healthcare settings.

## 1 Introduction

Large language models (LLM), such as ChatGPT and GPT-4, have showcased impressive capabilities in comprehending users' intent and generating coherent responses (Zhao et al., 2023). Their flexible input requirements make them suitable for a broad range of tasks across various domains. Among these, the medical domain stands out as a particularly fitting area for LLM applications (Thirunavukarasu et al., 2023). Over the past two decades, hospital IT systems like Hospital Information System (HIS), Laboratory Information System (LIS), and Picture Archiving and Communication System (PACS) have accumulated a wealth of clinical data, providing a robust foundation for training LLMs (Wu et al., 2024). Simultaneously, there is a significant demand for LLM applications in the medical field, such as online inquiries (Liu et al., 2022a), diagnostic assistance (Rasmy et al.,

---

[*]Corresponding author

2021), medication recommendations (Wu et al., 2022), and discharge summary (Liu et al., 2022c). Implementing LLMs in these medical scenarios can alleviate doctors' workload and enhance clinical efficiency (Wang et al., 2023b).

However, given the critical nature of patient care, the medical domain has little tolerance for errors in LLM outputs. Therefore, a comprehensive evaluation of these models is essential before deployment. Several benchmarks have been proposed to date, which can be categorized into three types: 1) Exam-based, CMExam (Liu et al., 2024b) is built from China National Medical Licensing Examination (CNMLE), MedQA (Jin et al., 2021) is built from United States Medical Licensing Examination (USMLE) and MedMCQA (Pal et al., 2022) is built from All India Institute of Medical Sciences (AIIMS PG) and National Eligibility cum Entrance Test (NEET PG). These licensing exams, designed to judge whether medical school students are qualified to be doctors, provide a reasonable basis for evaluating LLM performance; 2) QA-based, involving single (Abacha et al., 2017) and multi-turn interactions (Liu et al., 2022a) between patients and doctors, which can evaluate the performance of LLM in dealing with patients' inquiries; 3) Task-based, assessing performance in various medical Natural Language Processing (NLP) tasks, such as summarization, medical named entity recognition (NER), etc. For example, CBLUE (Zhang et al., 2022) is a collection of medical tasks. PromptCBLUE (Zhu et al., 2023) further adapts CBLUE for LLM evaluation by adding different forms of prompts. For more detailed information and illustrative examples, please refer to Appendix A.1.2.

A limitation of existing medical benchmarks is that they are organized either by the question type (multiple-choice, question answering) or the task type (NER, Classification, etc.), and many of them do not include clinical text data generated from real-world clinical practice (Wu et al., 2024). In addition, existing datasets are not structured according to the steps of the clinical process in patient care. As a result, it's difficult to assess the performance of LLMs in assisting patients in real clinics. In this paper, we segment the entire patient clinical journey into four stages. For each stage, we introduce multiple tasks. In summary, the contribution of this paper is threefold:

- We introduce a new Chinese benchmark dataset MedJourney [2] that covers the entire workflow of the patient's clinical journey which organizes the benchmark w.r.t patient clinical journey.

- In total, we introduce 12 datasets corresponding to 12 different tasks across four stages. Of these, 7 datasets are reconstructed from existing public sources, while 5 are newly proposed in this paper. All new datasets have been meticulously processed by professional doctors, ensuring high quality and reliability. The detailed information can be found in Appendix A.2.

- We evaluate the performance of existing LLMs on MedJourney. We evaluate not only the close source LLMs, like ChatGPT [3] and GPT-4 [4] but also open-source LLMs, like QWen (Bai et al., 2023) and ChatGLM (Du et al., 2022). We also include medical LLMs like Huatuo GPT2 (Chen et al., 2023b). Since MedJourney is in Chinese, we didn't include the English-centric LLMs, like Llama series (Touvron et al., 2023a,b). In addition to the accuracy and NLG metrics, we also conduct entity-level evaluations to verify performance from a semantic perspective.

## 2 Related Works

MultiMedQA (Singhal et al., 2023) is a widely recognized benchmark for evaluating the performance of LLMs in the medical domain. This benchmark comprises seven datasets: MedQA (Jin et al., 2021), MedMCQA (Pal et al., 2022), PubMedQA (Jin et al., 2019), MMUL clinical topics (Hendrycks et al., 2020), LiveQA (Abacha et al., 2017), MedicationQA (Abacha et al., 2019), and HealthSearchQA. The first four datasets consist of multiple-choice questions, while the last three contain questions requiring long-form free-text answers. This benchmark is in English. Recently, there are also some new clinical benchmarks that focus on diagnostic reasoning (Gao et al., 2023), multi-modal agent (Schmidgall et al., 2024) and doctor-patient conversation (Wang et al., 2023c).

For Chinese benchmarks, MedBench (Cai et al., 2024) primarily included multiple-choice questions from medical examinations. However, the MedBench website (Liu et al., 2024a) [5] featured a broader range of tasks, such as QA and NER. Liu et al. (2024b) constructed a benchmark, CMEExam, for

---

[2]https://github.com/Medical-AI-Learning/MedJourney
[3]https://openai.com/index/chatgpt/
[4]https://openai.com/index/gpt-4/
[5]https://medbench.opencompass.org.cn/home

Table 1: Comparison of Ours and Existing Medical Benchmarks.

| Benchmark | Journey Organized | Multiple Choice | QA | Summarization | Classification |
|---|---|---|---|---|---|
| MedExam | ✗ | ✗ | ✔ | ✗ | ✗ |
| MedBench (paper) | ✗ | ✔ | ✗ | ✗ | ✗ |
| MedBench (website) | ✗ | ✔ | ✔ | ✔ | ✔ |
| PromptCBLUE | ✗ | ✔ | ✔ | ✔ | ✔ |
| CMB | ✗ | ✗ | ✔ | ✔ | ✗ |
| MultiMedQA | ✗ | ✗ | ✔ | ✔ | ✗ |
| MedJourney (ours) | ✔ | ✔ | ✔ | ✔ | ✔ |

Figure 1: Four stages in patient journey which covers patients' experience workflow. For each stage, we introduce several datasets to evaluate the performance of LLMs.

evaluating LLMs based on medical exam data collected from the web. In addition to the answer, the authors appended five types of labels to each question, enabling a detailed performance analysis. CMB (Wang et al., 2023a) included both medical exams and some clinical case studies. Zhang et al. (2022) introduced CBLUE, which comprises multiple biomedical tasks. Since CBLUE was not specifically designed to evaluate LLMs' performance, Zhu et al. (2023) introduced multiple prompt templates for each task and restructured each instance in CBLUE into a prompt and target format. For free-form QA datasets, there are CMedQA (Zhang et al., 2017), CMedQA2 (Zhang et al., 2018), and WebMedQA (He et al., 2019). The differences between existing benchmarks and MedJourney are summarized in Table 1.

# 3 Patient Clinical Journey

Inspired by the ideal patient journey in the digital healthcare[6], as shown in Figure 1, we divide the patient clinical journey into four stages: planning, access, delivery, and ongoing care, and then propose 12 data sets accordingly, among which, Department Recommendation (DR), Pre-Consultation Dialogue Summary(PCDS), Hospital Reception QA (HQA), Insurance QA (IQA) and Drug QA (DQA) are newly proposed, while the rest seven are rebuilt from existing datasets. More details about how the patient clinical journey is divided can be found in Appendix A.1.1 and more details about how these data sets are built can be found in Appendix A.2.

## 3.1 Planning

At this stage, patients are becoming aware of potential health issues and are considering seeking medical care at a hospital. We propose three datasets to address the primary needs of patients at this stage: 1) Department Recommendation (DR) assists in identifying the most suitable departments based on patients' primary complaints; 2) Pre-Consultation Dialogue Summary (PCDS) utilizes a multi-turn conversation to gather and summarize patients' information; 3) Hospital Reception QA (HQA) compiles frequently asked questions from patients before and during their hospital visits.

### 3.1.1 Department Recommendation

As online appointment for medical consultations becomes increasingly popular, a growing number of patients are choosing to use the outpatient intelligent guidance system to select suitable hospitals and departments. Department recommendation is a crucial component of this system, aiming to suggest appropriate departments based on the patient's primary complaints. However, patients often lack adequate medical knowledge, resulting in vague symptom descriptions. Moreover, the complexity of hospital department structures further complicates the task of department recommendation.

---

[6]https://www.qualtrics.com/au/experience-management/industry/patient-journey/

Table 2: Examples of Department Recommendation. All texts used in this study are in Chinese. English translations are shown for reference.

| Patient complaint | Department |
|---|---|
| 我家孩子x型腿，同时膝盖内翻，想给他做腿型矫正，应该挂什么科？
My child has X-shaped legs and knee valgus. I would like to have leg alignment correction for him. Which department should I visit? | 小儿骨科，小儿关节外科，小儿矫形骨科
Pediatric Orthopedics, Pediatric Joint Surgery, Pediatric Orthopedic Correction |
| 容易失眠，晚上1-2点睡，白天睡不着，现有心慌心悸，感觉心脏不适
I have trouble falling asleep, going to bed around 1-2 am, being unable to sleep during the day, and experiencing palpitations and a sense of discomfort in the heart. | 心血管内科，神经内科，精神科
Cardiovascular Medicine, Neurology, Psychiatry |

To evaluate the effectiveness of LLMs in the task of department recommendation, we have compiled a Chinese dataset comprising 500 main complaints, each paired with a recommended department. In hospitals, departments are categorized with a high level of specificity. For example, the field of dentistry encompasses sub-specialties such as dental caries and endodontics, oral and maxillofacial surgery, dental implantology, orthodontics, and oral mucosal diseases. Moreover, the process of recommending departments for children significantly differs from that for adults.

Given the above considerations, we have chosen 100 fine-grained departments from 12 coarse-grained departments, which include 59 adult departments and 41 pediatric departments. Our selections were informed by the Directory of Medical Institutions' Clinical Departments[7] and real-world department structures. Table 2 shows two examples of patient complaints and the corresponding departments.

### 3.1.2 Pre-Consultation Dialogue Summary

To comprehend the patient's primary concerns, pre-consultation can commence with the patient's concise description of symptoms, with subsequent inquiries intelligently expanded based on the doctor's thought process. For instance, questions regarding the duration of symptoms, causative factors, characteristics, frequency, as well as inquiries about the patient's past medical history and allergy history. These pieces of information are then synthesized into a summary to assist doctors in obtaining an understanding of the patient's condition in advance.

Therefore, the pre-consultation dataset comprises multiple rounds of symptom-related conversations, culminating in a summary of that pre-consultation dialogue. To assess the capability of LLMs in the pre-consultation dialogue summarizing task, we have collected 100 conversation-summary pairs with an average of 19.84 rounds per conversation.

### 3.1.3 Hospital Reception QA

The dataset comprises a total of 133 entries, covering 10 common types of user queries in hospital visiting: Diagnostic Testing, Patient Care Process, Health Information, Healthcare Policy, Health Insurance, Departmental Information, Diagnostic Report, Department Referral, Wayfinding, Registration and Certification.

## 3.2 Access

Following the planning stage, the patient can consult with the appropriate doctor. During the face-to-face diagnosis, there are two primary tasks where LLM can be utilized: 1) Doctor Response Generation (DRG), which can generate the doctor's next response based on the historical patient-doctor dialogue. This can assist novice doctors and serve as a reminder for experienced doctors during the conversation; 2) Patient-Doctor Dialogue Summarization (PDDS), which can alleviate the workload associated with writing medical records by providing concise summaries.

### 3.2.1 Doctor Response Generation

Each medical dialogue consists of inquiries from the patient and responses from the physician, alternating in chronological order. Patients typically begin by describing their chief complaint and primary symptoms. Physicians need to further inquire about the patient's symptoms, disease history, and any examination results, guiding the conversation forward to better understand the patient's condition. Furthermore, physicians can provide helpful medical recommendations, including suggested examinations, preliminary diagnoses, and appropriate medications or treatments.

---

[7]http://www.nhc.gov.cn/fzs/s3576/201808/345269bd570b47e7aef9a60f5d17db97.shtml

To this end, we sampled 200 medical dialogues from the MedDG dataset (Liu et al., 2022b). The LLM is tasked with automatically generating a doctor's response based on the historical dialogue up to that moment. This dataset is derived from real-world online consultations, with professional physicians responding to patient inquiries. Table 3 shows examples of the DRG task.

Table 3: Example of Doctor Response Generation.

| Historical Dialogue | Response |
|---|---|
| 患者：偶尔恶心，胃疼，胆汁反流和胆囊炎有关吗？（男，23岁）

Patient: I occasionally experience nausea, stomach pain, and bile reflux. Could this be related to cholecystitis? (Male, 23 years old) | 医生：您好，有关系的，这种情况多久了？有吃药吗？
Doctor: Hello, yes, it could be related. How long have you been experiencing these symptoms? Did you take any medication? |
| 患者：面色黄，肚痛眼睛肿（男，44岁）
医生：您好，请问这种情况有多长时间了？
. . .
医生：肝功能异常。以前有肝病吗？平时喝酒多吗？
患者：平时喝酒挺多的。如果有变化的话就这几年。单从这上边看最大可能是什么呢。
医生：这上面看就是肝功能异常，其他看不出来，要是以前没有乙肝，在你喝酒多就是酒精性高损伤。
患者：好的。需要做什么进一步检查呢？
Patient: I have yellowish skin, stomach pain, and swollen eyes. (Male, 44 years old)
Doctor: Hello, how long have you been experiencing these symptoms?
. . .
Doctor: Abnormal liver function. Have you had any liver disease in the past? Do you drink frequently?
Patient: I often drink. If there has been any change, it's been in the past few years. Based on these symptoms, what is the most likely diagnosis?
Doctor: It appears to be abnormal liver function. If you haven't had hepatitis B before and you drink a lot, it is likely alcohol-related liver damage.
Patient: Okay. What further test should I undergo? | 医生：查个腹部彩超，如果没查过乙肝三项，也检查一下。




Doctor: Get an abdominal ultrasound. If you haven't checked for the hepatitis B markers, check it too. |

### 3.2.2 Doctor Patient Dialogue Summarization

After the consultation, the physician needs to summarize the dialogue into a brief medical report. The report typically consists of six sections: chief complaint, history of present disease, auxiliary examinations, history of past disease, diagnosis, and advice. To this end, we sampled 200 real-world medical dialogues and the corresponding summaries from the IMCS21 dataset (Chen et al., 2023a). The LLM is tasked with automatically summarizing a brief report based on the medical dialogue.

### 3.3 Delivery

Upon receiving a patient's information, doctors carefully review it to assess the individual's medical history, current symptoms, and any pertinent diagnostic test results. They are then tasked with providing an initial diagnosis. This process entails determining the necessity for further examinations, identifying the patient's disease, and deciding on the most suitable medication and treatment for their condition. To evaluate whether the medical LLM can accurately assess a patient's complex condition in real-world clinical diagnoses, we've divided the service delivery into four stages: predicting examinations, predicting diseases, predicting treatments, and predicting medications. We've developed datasets for each of these components. These four datasets are constructed from the CMB (Wang et al., 2023a) and CMExam (Liu et al., 2024b) datasets.

### 3.3.1 Examination Prediction

After developing a preliminary understanding of the patient's condition, the doctor proceeds to explore various potential diagnoses. To refine these possibilities or to rule out specific conditions, the doctor should determine the appropriate examinations needed to either confirm their initial assessment or delve deeper into their investigation. These tests could range from blood work and imaging studies (such as X-rays or MRIs) to biopsies or other specialized exams tailored to the patient's symptoms or suspected condition. Therefore, it's essential to assess whether a medical LLM can fully comprehend the patient's situation and recommend the most appropriate examination. The examination prediction dataset consists of 432 question-answer pairs, encompassing a total of 360 examinations. Examples of examination prediction question-answer pairs can be found in Table 4.

### 3.3.2 Disease Prediction

After carefully reviewing all the available information, the doctor comes to a diagnosis. This important step requires the doctor to use their wide-ranging medical knowledge and expertise. In the same way,

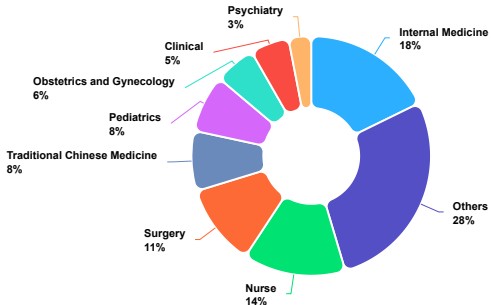

Figure 2: The distribution of disease subjects.

Table 4: Example of Examination Prediction.

| Question | Answer |
| --- | --- |
| 女34岁。月经量进行性减少，现闭经半年，泌乳3个月，首选检查项目应是？可能的检查包括：孕激素试验、血HCG测定、血PRL测定、性激素测定、诊断性刮宫。 | 血PRL测定 |
| Woman, 34 years old. Her menstrual flow has been progressively decreasing, and she has now been amenorrheic for six months, with lactation for three months. What should be the first choice of examination? Possible tests include: pregnancy hormone test, blood HCG measurement, blood PRL measurement, sex hormone measurement, diagnostic curettage. | Blood PRL measurement |

a skilled medical LLM should also understand the small differences and details that can separate similar diseases or conditions. They must be good at recognizing patterns, thinking about different possibilities, and judging the chance of each possible diagnosis based on the evidence they have. This dataset includes 1,761 question-answer pairs that cover a total of 1,490 diseases.

### 3.3.3 Treatment Prediction

Doctors typically rely on established medical guidelines and evidence-based medicine when devising treatment plans. With all the relevant information at their disposal, they tailor the treatment plan to suit the patient's unique needs. The treatment prediction dataset serves as a tool to assess whether medical LLMs can generate accurate treatment plans that not only address the patient's diagnosis but also take into account their individual circumstances. This dataset comprises a total of 148 question-answer pairs, encompassing 133 distinct treatment plans.

### 3.3.4 Medication Prediction

When recommending medications to patients, doctors consider the effectiveness, safety, potential side effects, and dosage requirements of the drugs. Prescribing an incorrect medication can lead to serious repercussions, such as not addressing the actual issue, potentially exacerbating the patient's condition, or postponing the necessary treatment. Therefore, it's essential for medical LLMs to possess a deep understanding of medicine and provide the most precise medication recommendations tailored to a patient's unique situation. The medicine prediction dataset comprises a total of 1029 question-answer pairs, covering 722 distinct medications.

## 3.4 Ongoing Care

At this stage, patients have transitioned from receiving in-hospital care to managing their health independently outside the hospital. There are three key tasks where LLM can be applied: 1) Drug QA (DQA), this task involves addressing common questions patients may have about their medications; 2) Insurance QA (IQA), this task includes answering frequently asked questions about medical insurance; 3) Mental Health QA (MQA), this task involves addressing common questions from patients who may be dealing with mental health issues outside the hospital.

### 3.4.1 Drug QA

Drug QA is a critical component in the post-treatment phase of a patient's clinical journey, where patients often seek clarity on their prescribed medications. Understanding the indications, contraindications, side effects, dosage instructions, and potential interactions of medications is essential for safe and effective self-care. To address this need, we have curated a dataset comprising 137 question-answer pairs that reflect real-world patient inquiries about various medications. Table 5 provides two examples of medication question-answer pairs.

### 3.4.2 Insurance QA

Medical insurance is another important aspect of the healthcare system, especially after the patients are discharged from the hospital. The patients need to pay for the medical expenses, and the medical

Table 5: Examples of Drug QA.

| Query | Answer |
|---|---|
| 卡马西平片的副作用？

What are the side effects of Carbamazepine? | 卡马西平片常见不良反应主要包括，眩晕症，嗜睡，乏力恶心，呕吐等消化道症状，也可以引起骨髓抑制，中毒性肝炎，更为罕见的是可以引起剥脱性皮炎。
Common adverse reactions to Carbamazepine mainly include dizziness, drowsiness, fatigue, nausea, and other gastrointestinal symptoms. It can also cause bone marrow suppression, toxic hepatitis, and, more rarely, exfoliative dermatitis. |
| 孕妇能吃维生素E吗？
Can pregnant women take vitamin E? | 孕妇可以吃维生素E，维生素E是一种脂溶性维生素，又称为生育酚，主要是抗氧化作用。
Pregnant women can take vitamin E. Vitamin E is a fat-soluble vitamin, also known as tocopherol, and it primarily has antioxidant properties. |

insurance can help them reduce the financial burden. However, the medical insurance policies are usually complex and difficult to understand. The patients may have many questions about the medical insurance, such as what drugs are covered by the insurance and how to claim the insurance. The answers are provided by the medical insurance experts.

### 3.4.3 Mental Health QA

Many patients may experience mental health issues after receiving treatment or hospitalization. This is partly due to the disease itself, which significantly impacts both the physical and mental health of patients. Additionally, long-term treatment and hospitalization, side effects from medications, and lengthy post-surgery recovery all impose considerable psychological burdens on patients and their families. Consequently, many of them face psychological challenges post-treatment, necessitating mental health support. To address this need, we incorporated a Mental Health QA (MQA) task to aid patients. Therefore, we sampled 200 MQA instances from the PsyQA dataset (Sun et al., 2021), consisting of mental health inquiries from patients and responses from professionals.

## 4 Performance of LLM on the Benchmark

### 4.1 Experiment Setting

We have chosen three types of LLMs for evaluation: 1) Open-source LLMs, such as ChatGLM3 (Du et al., 2022) and QWen 1.5 (Bai et al., 2023), with parameter sizes ranging from 7b to 72b; 2) Closed-source LLMs, like ChatGPT and GPT-4, for which we are using the API of version 1106; 3) Medical domain-specialized LLMs, like HuatuoGPT2 (Chen et al., 2023b) and DISC-MedLLM (Bao et al., 2023). Since the proposed benchmark is in Chinese, we primarily select LLMs that are well-trained on Chinese corpora and exclude English-centric LLMs like LLama (AI@Meta, 2024). The inference is conducted on a server with 8 NVidia A100.

For metrics, for classification tasks such as Department Recommendation (DR) and multiple-choice tasks like Examination Prediction (EP), Disease Prediction (DP), Treatment Prediction (TP), and Medication Prediction (MP), we use accuracy as the metric; For summarization tasks, like Patient Doctor Dialogue Summarization (PDDS), and generation tasks, like Hospital Reception QA (HQA), Doctor Response Generation (DRG), Drug QA (DQA), Insurance QA (IQA), and Mental Health QA (MQA), we use Natural Language Generation (NLG) metrics, such as BLEU (Papineni et al., 2002). In order to focus on the key medical entities, we introduce entity-based metrics. In addition to automatic metrics, we also include LLM evaluation and human evaluation.

### 4.2 Experimental Results

#### 4.2.1 Basic Model

Table 6 presents the zero-shot performance of various LLMs on the proposed MedJourney. It appears that no single LLM excels across the entire patient clinical journey. Only Huatuo2-34B surpasses other models in the Delivery Stage, suggesting that it's likely specifically optimized for those tasks.

#### 4.2.2 One-Shot Performance

In addition to the zero-shot setting, we also explore a few-shot setting. Given that the average prompt length for some tasks, such as PDDS, can be quite lengthy and exceed the token limit of smaller models, we opt for a one-shot approach in this experiment. As shown in Table 7, for general models

Table 6: Zero-Shot Performance on the Clinical Journey Dataset.

| Model | DR | PCDS | HQA | DRG | PDDS | EP | DP | TP | MP | DQA | IQA | MQA |
|---|---|---|---|---|---|---|---|---|---|---|---|---|
| | Acc | B-4 | B-4 | B-4 | B-4 | Acc | Acc | Acc | Acc | B-4 | B-4 | B-4 |
| *Public LLMs* | | | | | | | | | | | | |
| ChatGLM3 | 0.130 | 9.828 | 2.462 | 2.138 | 6.818 | 0.398 | 0.313 | 0.331 | 0.318 | 7.108 | **5.323** | 3.740 |
| QWen-7B | 0.264 | 5.489 | 1.843 | 1.773 | 7.796 | 0.528 | 0.559 | 0.493 | 0.638 | 7.256 | 4.241 | 3.917 |
| QWen-14B | 0.280 | 8.271 | 2.151 | **3.740** | 6.560 | 0.630 | 0.579 | 0.588 | 0.646 | 8.296 | 4.993 | 4.071 |
| QWen-32B | 0.304 | 7.614 | 1.978 | 2.491 | 6.987 | 0.667 | 0.684 | 0.608 | 0.715 | **8.596** | 4.609 | **4.214** |
| QWen-72B | **0.308** | 9.032 | 2.266 | 2.649 | 9.318 | 0.674 | 0.692 | 0.635 | 0.742 | 8.371 | 4.671 | 4.129 |
| *Private LLMs* | | | | | | | | | | | | |
| ChatGPT | 0.255 | **11.242** | **2.576** | 2.912 | 10.632 | 0.419 | 0.369 | 0.364 | 0.382 | 6.928 | 5.303 | 3.767 |
| GPT-4 | 0.306 | 5.913 | 1.485 | 1.649 | 7.453 | 0.613 | 0.632 | 0.503 | 0.561 | 6.623 | 3.694 | 3.598 |
| *Specialized LLMs* | | | | | | | | | | | | |
| HuatuoGPT2-7B | 0.214 | 4.387 | 2.094 | 1.410 | 6.200 | 0.681 | 0.688 | 0.554 | 0.629 | 8.079 | 4.165 | 4.035 |
| HuatuoGPT2-34B | 0.278 | 7.951 | 1.764 | 1.998 | **12.082** | **0.757** | **0.766** | **0.662** | **0.777** | 8.382 | 4.149 | 4.132 |
| DISC-MedLLM | 0.216 | 2.775 | 2.524 | 1.288 | 6.190 | 0.269 | 0.241 | 0.149 | 0.282 | 5.570 | 3.928 | 2.908 |

Table 7: One-Shot Performance on the Clinical Journey Dataset.

| Model | DR | PCDS | HQA | DRG | PDDS | EP | DP | TP | MP | DQA | IQA | MQA |
|---|---|---|---|---|---|---|---|---|---|---|---|---|
| | Acc | B-4 | B-4 | B-4 | B-4 | Acc | Acc | Acc | Acc | B-4 | B-4 | B-4 |
| *Public LLMs* | | | | | | | | | | | | |
| ChatGLM3 | 0.258 | 8.282 | **4.715** | 2.555 | 20.674 | 0.465 | 0.279 | 0.480 | 0.343 | 6.987 | **6.432** | 3.661 |
| QWen-7B | 0.292 | 10.414 | 2.386 | 2.410 | 12.868 | 0.412 | 0.436 | 0.365 | 0.485 | 7.207 | 5.129 | 4.074 |
| QWen-14B | 0.392 | 14.554 | 2.515 | 3.361 | 11.591 | 0.539 | 0.583 | 0.520 | 0.580 | 7.724 | 5.538 | 3.935 |
| QWen-32B | 0.354 | 12.567 | 2.349 | **4.305** | 13.788 | **0.676** | 0.739 | 0.709 | 0.699 | 7.971 | 5.030 | **4.570** |
| QWen-72B | 0.370 | 12.545 | 2.713 | 2.918 | 14.946 | 0.674 | **0.769** | **0.730** | **0.723** | **8.906** | 5.532 | 4.438 |
| *Private LLMs* | | | | | | | | | | | | |
| ChatGPT | 0.328 | **21.316** | 4.437 | 3.199 | 18.714 | 0.364 | 0.345 | 0.357 | 0.355 | 5.915 | 6.091 | 3.911 |
| GPT-4 | **0.440** | 13.164 | 2.506 | 2.169 | 12.834 | 0.601 | 0.679 | 0.575 | 0.576 | 7.817 | 3.886 | 4.022 |
| *Specialized LLMs* | | | | | | | | | | | | |
| HuatuoGPT2-7B | 0.208 | 2.182 | 1.674 | 1.526 | 5.224 | 0.410 | 0.399 | 0.250 | 0.266 | 8.178 | 4.545 | 4.061 |
| HuatuoGPT2-34B | 0.274 | 12.159 | 2.567 | 2.958 | 16.244 | 0.641 | 0.588 | 0.466 | 0.659 | 8.569 | 5.248 | 4.320 |
| DISC-MedLLM | 0.212 | 0.831 | 3.187 | 1.856 | 9.553 | 0.065 | 0.163 | 0.101 | 0.244 | 3.943 | 5.072 | 2.984 |

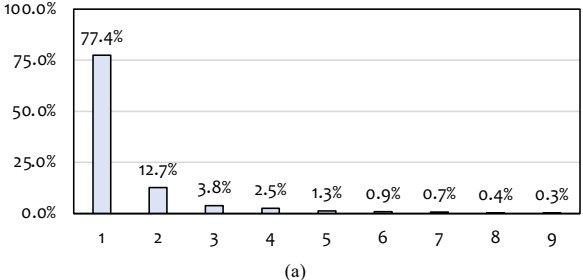

| Rank | Entity | Translation |
|---|---|---|
| 1 | 咳嗽 | Cough |
| 2 | 发热 | Fever |
| 3 | 腹泻 | diarrhea |
| 4 | 血常规 | complete blood count |
| 5 | 奥美拉唑 | Omeprazole |

(a)                    (b)

Figure 3: (a) Number of unique entities sorted by frequency. (b) Top ranked annotated entities.

like GPT-4 and QWen, incorporating a one-shot approach enhances performance across most tasks. This improvement is particularly noticeable for the summary tasks PDDS, as the LLMs learn the format of the output content. However, for delivery tasks, the performance of medical domain-specific models decreases. The introduced example may act as noise, negatively impacting performance.

### 4.2.3 Entity Level Evaluation

For QA tasks, we incorporate an entity-based metric alongside general NLG metrics to evaluate LLMs from a semantic perspective. Specifically, we extract medical entities such as disease names, symptom terms, drug names, etc., from the golden answer of each instance. This extraction is initially performed by GPT-4 and subsequently double-checked manually. The frequency of these extracted entities is depicted in Figure 3(a). The 5 most frequent entities are listed in Figure 3(b).

Using these extracted entities, we compute the entity recall for all QA tasks. As demonstrated in Table 8, GPT-4 achieves near-optimal performance on almost all tasks, suggesting that GPT-4 is capable of incorporating key information in its responses.

Table 8: The Recall of Medical Entities for QA Tasks.

| Model | PCDS | HQA | DRG | PDDS | DQA | IQA | MQA |
|-------|------|-----|-----|------|-----|-----|-----|
| *Public LLMs* | | | | | | | |
| ChatGLM3 | 0.461 | 0.227 | 0.132 | 0.381 | 0.151 | 0.287 | 0.220 |
| QWen-7B | 0.495 | 0.281 | 0.155 | 0.428 | 0.188 | 0.321 | **0.287** |
| QWen-14B | 0.513 | 0.262 | 0.096 | **0.428** | **0.192** | 0.326 | 0.286 |
| QWen-32B | 0.524 | 0.270 | 0.175 | 0.420 | 0.176 | 0.340 | 0.259 |
| QWen-72B | 0.495 | **0.288** | 0.182 | 0.417 | 0.188 | 0.324 | 0.255 |
| *Private LLMs* | | | | | | | |
| ChatGPT | 0.462 | 0.250 | 0.151 | 0.363 | 0.148 | 0.302 | 0.247 |
| GPT-4 | **0.603** | 0.274 | **0.184** | 0.400 | 0.184 | **0.341** | 0.271 |
| *Specialized LLMs* | | | | | | | |
| HuatuoGPT2-7B | 0.415 | 0.234 | 0.151 | 0.327 | 0.184 | 0.309 | 0.241 |
| HuatuoGPT2-34B | 0.441 | 0.239 | 0.142 | 0.357 | 0.194 | 0.314 | 0.265 |
| DISC-MedLLM | 0.312 | 0.212 | 0.157 | 0.274 | 0.126 | 0.211 | 0.180 |

### 4.2.4 LLM Rating

We also leverage the LLMs to evaluate the performance of LLMs. We let GPT-4 to rate the performance of LLMs on benchmark, paticulary for the QA questions.

---

**The Prompt for GPT-4 to Rate the Performance of QA Tasks**

Below is a medical task with responses from both the large language model and the ground-truth provided by human annotation. Based on the 3 criteria below, rate the model's performance on a scale of 1-100. Only provide the scores without explanations.

**Accuracy:** The response provided by the LLM is accurate and has no factual errors. Conclusions not made arbitrarily.
**Helpfulness:** The model's response provides the patient with clear, instructive and practical assistance, specifically addressing the medical task.
**Linguistic Quality:** The response logical. The model correctly understands the medical task, and the expressions smooth and natural.

Please ensure that you do not let the length of the text influence your judgment, do not have a preference for any AI assistant names that might appear in the dialogue, do not let irrelevant linguistic habits in the conversation influence your judgment, and strive to remain objective.Your scoring should be strict enough and do not give a perfect score easily.

---

The text box above displays the prompt that instructs GPT-4 to evaluate the performance of LLMs on the QA tasks in MedJourney. As shown in Table 9, if we average the performance on these 7 tasks. QWen-72B, GPT-4 and QWen 32b rank the top 3.

### 4.2.5 Human Evaluation

In addition to the automatic metrics, we have also carried out a human evaluation of 10 Language Learning Models (LLMs) across 7 QA tasks. For this evaluation, we instructed the annotators to assign a score from 1 to 100, based on the following criteria. Table 10 displays the average human rating of performance of LLMs on each task.

---

**The Instruction of Human Evaluation**

Based on the 3 criteria below, rate the model performance on a scale of 1-100.

**Accuracy:** The response provided by the large language model is accurate and has no factual errors. Conclusions are not made arbitrarily.
**Helpfulness:** The model's response provides the patient with clear, instructive and practical assistance, specifically addressing the medical task.
**Linguistic Quality:** The response logical. The model correctly understands the medical task, and the expressionis smooth and natural.

---

In this analysis, we average the performance across various QA tasks. GPT-4 emerges as the top performer, followed by QWen-32B, which deviates from the LLM rating ranking presented in Section 4.2.4. This raises the question of which metric aligns best with human evaluation: the B-4 metric (Table 6), the entity-based metric (Table 8), or the LLM rating (Table 9). We rank 10 Language Learning Models (LLMs) based on the average metrics across the seven tasks, as shown in Table 11.

To calculate the alignment between each metric and human evaluation, we enumerate all pairs of the 10 models, resulting in a total of 45 pairs. If the order of a pair of models according to one metric is

Table 9: The GPT-4 Evaluation on QA Tasks in MedJourney.

| Model | PCDS | HQA | DRG | PDDS | DQA | IQA | MQA | AVERAGE |
|---|---|---|---|---|---|---|---|---|
| *Public LLMs* | | | | | | | | |
| ChatGLM3 | 80.16 | 78.34 | 81.83 | 82.35 | 82.66 | 82.41 | 87.96 | 82.24 |
| QWen-7B | 87.37 | 83.32 | 85.83 | 86.67 | 87.32 | **87.53** | **90.30** | 86.91 |
| QWen-14B | 89.37 | 85.58 | 81.69 | 87.06 | 87.68 | 86.09 | 89.44 | 86.70 |
| QWen-32B | 88.24 | 85.80 | **86.22** | 87.32 | 88.13 | **87.53** | 90.08 | 87.62 |
| QWen-72B | 89.34 | **86.57** | 85.64 | **87.84** | **88.88** | 86.42 | 89.45 | **87.73** |
| *Private LLMs* | | | | | | | | |
| GhatGPT | 88.18 | 83.10 | 69.95 | 86.05 | 83.43 | 84.06 | 89.89 | 83.52 |
| GPT-4 | **89.44** | 85.86 | 85.86 | 87.51 | 87.80 | 87.33 | 90.19 | 87.71 |
| *Specialized LLMs* | | | | | | | | |
| HuatuoGPT2-7B | 81.89 | 79.41 | 82.94 | 81.41 | 87.24 | 84.99 | 88.61 | 83.78 |
| HuatuoGPT2-34B | 89.11 | 82.44 | 84.58 | 85.75 | 86.34 | 86.28 | 89.38 | 86.27 |
| DISC-MedLLM | 72.45 | 78.89 | 81.83 | 77.69 | 79.37 | 77.01 | 86.31 | 79.08 |

Table 10: The Human Evaluation on QA Tasks in MedJourney.

| Model | PCDS | HQA | DRG | PDDS | DQA | IQA | MQA | AVERAGE |
|---|---|---|---|---|---|---|---|---|
| *Public LLMs* | | | | | | | | |
| ChatGLM3 | 87 | 46 | 61 | 83 | 75 | 37 | 72 | 66 |
| QWen-7B | 96 | 65 | 70 | 87 | 78 | 53 | 73 | 75 |
| QWen-14B | 97 | 64 | 61 | 88 | 77 | 56 | 73 | 74 |
| QWen-32B | 96 | 65 | 74 | 87 | **79** | 62 | 70 | 76 |
| QWen-72B | 97 | 62 | 73 | 86 | **79** | 51 | 72 | 74 |
| *Private LLMs* | | | | | | | | |
| ChatGPT | 94 | 59 | 66 | 80 | 75 | 46 | 79 | 71 |
| GPT-4 | **98** | **68** | **76** | **91** | 78 | **64** | **82** | **80** |
| *Specialized LLMs* | | | | | | | | |
| HuatuoGPT2-7B | 90 | 56 | 68 | 76 | 78 | 49 | 70 | 70 |
| huatuoGPT2-34B | 96 | 61 | 74 | 86 | **79** | 58 | 73 | 75 |
| DISC-MedLLM | 68 | 42 | 67 | 80 | 71 | 39 | 64 | 62 |

Table 11: The Ranking of 10 LLMs w.r.t Each Evalution Metric.

| RANK | B-4 | Entity | GPT Judge | GPT Judge + Entity | Human |
|---|---|---|---|---|---|
| 1 | ChatGPT | GPT-4 | QWen-72B | GPT-4 | GPT-4 |
| 2 | HuatuoGPT2-34B | QWen-32B | GPT-4 | QWen-32B | QWen-32B |
| 3 | QWen-72B | QWen-7B | QWen-32B | QWen-72B | HuatuoGPT2-34B |
| 4 | QWen-14B | QWen-72B | QWen-7B | QWen-7B | QWen-7B |
| 5 | ChatGLM3 | QWen-14B | QWen-14B | QWen-14B | QWen-72B |
| 6 | QWen-32B | HuatuoGPT2-34B | HuatuoGPT2-34B | HuatuoGPT2-34B | QWen-14B |
| 7 | QWen-7B | ChatGPT | HuatuoGPT2-7B | ChatGPT | ChatGPT |
| 8 | GPT-4 | HuatuoGPT2-7B | ChatGPT | HuatuoGPT2-7B | HuatuoGPT2-7B |
| 9 | HuatuoGPT2-7B | ChatGLM3 | ChatGLM3 | ChatGLM3 | ChatGLM3 |
| 10 | DISC-MedLLM | DISC-MedLLM | DISC-MedLLM | DISC-MedLLM | DISC-MedLLM |

the same as that in human evaluation, we assign it a vote of 1; otherwise, it receives a vote of 0. We then use the alignment rate to measure the corelation of between a metric and human evaluation.

When comparing the rankings of each pair of LLMs to see if they align with human ratings, we find that the correlation between Human and GPT Judge is 84.4%; between Human and Entity, it's 73.3%; and between Human and B-4, it's 35.5%. However, when we combine the scores from GPT-Judge and Entity (calculated as GPT-Judge score/100 + Entity score), the alignment improves to 91.1%. This combined metric offers a robust measure for evaluating the performance of a Language Learning Model in the medical domain.

## 5 Conclusion

In this paper, we introduce a novel benchmark dubbed MedJourney, which is designed to evaluate the performance of LLMs from the perspective of a patient's clinical journey. We segment the entire journey into four stages, and for each stage, we propose multiple tasks that LLMs can undertake, providing corresponding test sets for evaluation. Of the 12 datasets proposed, seven are constructed from existing corpora, while the remaining five are newly proposed datasets. In addition to the prompt and target pairs, we also provide the key medical entities that needs to be addressed. The proposed MedJourney allows us to assess LLMs from a clinical perspective and identify which stages require further improvement.

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

# A Appendix

## A.1 Background

### A.1.1 Stages in the Patient Clinical Journey

In this paper, we segment the clinical process based on the patient's journey, specifically focusing on the stages before visiting the hospital (planning), during the hospital visit (access and delivery), and after the hospital visit (ongoing care). This segmentation aligns with similar divisions proposed in previous works.

In Sehrawat (2023), the authors segment the patient journey into six stages: 1) Initial Contact and Symptom Recognition; 2) Diagnostic Evaluation and Treatment Planning; 3) Treatment and Care Delivery; 4) Follow-up Care and Monitoring; 5) Long-term Management and Disease Prevention; 6) Patient Education and Empowerment. Here, stage 1 aligns with our "planning" stage, stage 2 with "access", stage 3 with "delivery", and stages 4 to 6 with "ongoing care".

In Donaldson et al. (2021), the authors define the patient journey as "encounters with healthcare facilities, a hospital unit, a specialist visit, a primary care clinic, a home health agency". In this definition, "healthcare facilities" corresponds to our planning stage, "hospital unit" and "specialist visit" align with the access and delivery stages, and "primary care clinic" and "home health agency" map to the ongoing care stage.

In Gualandi et al. (2019), the authors describe a patient's journey through surgery, which includes seven stages: 1) Out-patient visit, 2) Examination at out-patient clinics, 3) Hospitalization and surgery, 4) Post-surgical care, 5) Discharge, 6) Rehabilitation Stay, 7) Follow-up visit. Here, stage 1 corresponds to "planning", stage 2 to "access", stages 3 and 4 to "delivery", and stages 5 to 7 to "ongoing care".

During the planning stage, patients experiencing symptoms are guided to the appropriate department through Department Recommendation (DR). The Pre-Consultation Dialogue Summary (PCDS) collects the patient's main complaints in a dialog format and summarizes them for the doctor's review. The Hospital Reception QA (HQA) informs patients about the specifics of their hospital visit, such as what items to bring, dietary restrictions before certain examinations, and so on.

In the access stage, the Doctor Response Generation (DRG) provides potential responses to the doctor based on the patient-doctor conversation. The Patient Doctor Dialogue Summarization (PDDS) further condenses the dialogue for easy reference.

During the delivery stage, Examination Prediction (EP), Disease Prediction (DP), Treatment Prediction (TP), and Medication Prediction (MP) recommend potential actions for the doctor's reference, such as suggesting further examinations, possible diagnoses, appropriate treatments, and medications for the patient.

In the ongoing care stage, Drug QA (DQA) instructs patients on medication usage and related knowledge. Insurance QA (IQA) provides information on medical insurance, and Mental Health QA (MQA) focuses on the patient's mental well-being.

While the proposed 12 tasks do not cover all aspects of a clinical process, such as radiology report generation, they effectively connect the dots of a patient's journey.

### A.1.2 Categories of Medical Tasks

Typical medical tasks can be categorized into three types: 1) Exam-based, 2) QA-based, 3) Task-based.

For the Exam-based tasks, questions are typically selected from the United States Medical Licensing Examination (USMLE) or the China National Medical Licensing Examination (CNMLE). These questions usually come in the form of a problem, multiple-choice options, and an answer.

For instance:

**Question:** A 77-year-old male presents with progressive right-hand tremors and slow movements. The patient has a history of benign prostatic hyperplasia and mild renal insufficiency. Which medication

would be most appropriate for his treatment? Candidate Options: (A) Artane (B) Levodopa (C) Selegiline (D) Amantadine (E) Bromocriptine

**Answer:** (B) Levodopa

For the QA-based tasks, an example can be found below.

**Question:** What are the side effects of Carbamazepine?

**Answer:** Common adverse reactions to Carbamazepine primarily include dizziness, drowsiness, fatigue, nausea, and other gastrointestinal symptoms. It can also cause bone marrow suppression, toxic hepatitis, and, more rarely, exfoliative dermatitis.

For Task-based tasks, one example involves extracting named entities from plain text.

**Prompt:** Please identify the medical name entity in this sentence, "Tetanus Spasm toxin has a long-term effect on the autonomic nerve."

**Target:** Autonomic nerve; Tetanus spasm toxin

## A.2   Dataset Description

In this paper, we present 12 data sets to cover the entire patient journey, among which, Department Recommendation (DR), Pre-Consultation Dialogue Summary (PCDS), Hospital Reception QA (HQA), Insurance QA (IQA) and Drug QA (DQA) are newly proposed, while the rest are rebuilt from existing datasets. These five tasks have corresponding applications in Tencent Healthcare Smart Hospital Solutions [8]. Based on user behaviours and medical knowledge, profession clinicians generated these 5 data sets either by rewriting existing cases or summarizing common enqueries. Among these 5 data sets, Hospital Reception QA (HQA), Drug QA (DQA), and Insurance QA (IQA) primarily focus on medical knowledge and contain minimal personal information.

The Department Recommendation (DR) and Pre-Consultation Dialogue (PCD) datasets are derived from users' inquiries prior to doctor visits. Therefore, these datasets focus on patients' primary complaints and do not include in-hospital data like examination, diagnosis, or medication information. In addition, we also employ experienced physicians to recompose the data instead of directly using patients' input, further ensuring the absence of patient information.

### A.2.1   Department Recommendation

Through the Department Recommendation service, a patient provides a description of their symptoms, and the service suggests the most appropriate department for the patient to visit.

We enlisted the help of three doctors for this annotation process, one of whom acted as the meta-annotator. In the first step, each department was initially assigned to a doctor who composed more than five patient complaints specific to that department. In the second step, another doctor supplemented potential departments based on the patient complaints generated in the first step. Finally, the meta-annotator evaluated each case, retaining the top five cases of the highest quality for each department. Since we target 100 departments, this process culminated in a final dataset comprising 500 entries. Figure 4 illustrates the distribution of the departments.

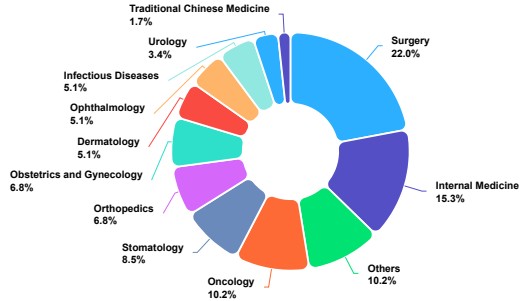

Figure 4: The distribution of coarse-grained departments.

---

[8]https://healthcare.tencent.com/solution/7

### A.2.2 Pre-Consultation Dialogue Summary

For the PCD dataset, in order to construct a representative dataset, we have collected 77 different symptoms from 36 fine-grained departments based on real user consultations. The representative departments and symptoms include:

- Oral Department (19%, including oral restoration, orthodontics, etc.): loose teeth, oral ulcers, impacted teeth, etc.,
- Dermatology Department (13%): skin itching, rashes, pigmentation, etc.,
- Pediatrics Department (12%): abnormal height, early loss of deciduous teeth, spitting up milk, cough, etc.,
- Ophthalmology Department (8%): decreased vision, trichiasis, ptosis, etc.,
- Gynecology Department (7%): infertility, irregular menstruation, abnormal vaginal discharge, etc.
- Respiratory Medicine Department (5%): asthma, dyspnea, upper respiratory tract infection, etc.

Table 12 provides an example of a pre-consultation conversation and the corresponding summary in this dataset. The summary is generated using the conversation as input.

Table 12: Example of Pre-Consultation Dialogue.

| Pre-Consultation dialogue | Summary |
|---|---|
| 患者: 牙齿松动
系统: 请问牙齿松动发生多久了?
患者: 2月
系统: 请问松动牙齿的数量是什么? 1个,2个及以
患者: 松动2颗或以上
系统: 请问牙齿松动的位置是右上后牙,上前牙,左上后牙,右下后牙,下前牙,左下后牙?
患者: 右上后牙，上前牙，左上后牙
系统: 请问牙齿松动是否慢慢加重? 患者: 不清楚
系统: 请问在以下哪些情况下发生牙齿松动? 外伤,进食硬物,夜里睡觉磨牙,其他
患者: 以上都没有
系统: 是否使用过以下药物 青霉素类,磺胺类,头孢菌素类,阿司匹林,布洛芬,感冒药,其它药物,无?
患者: 使用过头孢菌素类药物 | 患者2月前出现右上后牙、上前牙、左上后牙

牙齿松动，松动2颗或以上。使用过头孢菌素类药物。 |
| Patient: My teeth are loose.
Agent: How long has the tooth loosening been occurring?
Patient: 2 months.
Agent: How many teeth are loose? One, two, or more?
Patient: Two or more.
Agent: Can you specify the location of the loose teeth? Is it the upper right back teeth, upper front teeth, upper left back teeth, lower right back teeth, lower front teeth, or lower left back teeth?
Patient: Upper right back teeth, upper front teeth, and upper left back teeth.
Agent: Has the tooth loosening gradually worsened?
Patient: Not sure.
Agent: In which of the following situations does the tooth loosening occur? Trauma, eating hard foods, grinding teeth at night, other?
Patient: None of the above.
Agent: Have you used any of the following medications (multiple choices)? Penicillins, sulfonamides, cephalosporins, aspirin, ibuprofen, cold medicine, other drugs, none?
Patient: I have used cephalosporin drugs. | Patient experienced tooth loosening in the upper right back teeth, upper front teeth, and upper

left back teeth 2 months ago, with two or

two or more teeth being loose. The patient has used cephalosporin drugs.
cephalosporin drugs. |

### A.2.3 Hospital Reception QA

Table 13 showcases examples of queries related to the category of "Hospitalization Procedures". These examples provide insights into the type of questions users typically ask and how the AI bot responds, thereby serving as a valuable resource for improving the bot's performance. The distribution of data across these categories is illustrated in Figure 5.

### A.2.4 Doctor Patient Dialogue Summarization

Table 14 shows a case of Doctor Patient Dialogue Summarization.

### A.2.5 Disease Prediction

Examples of disease prediction question-answer pairs are in Table 15.

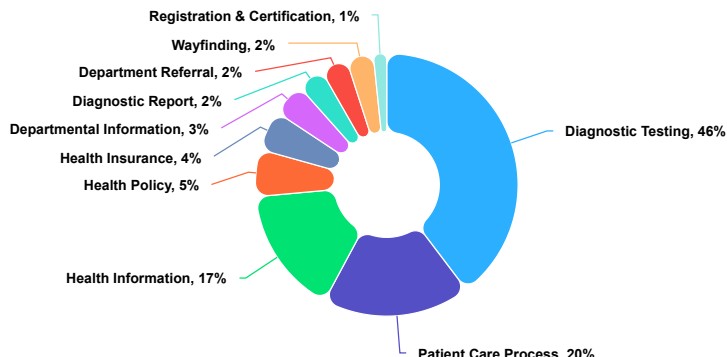

Figure 5: The distribution of intents of Hospital Reception QA.

Table 13: Two examples of queries about hospitalization procedures. English translations are shown for reference.

| Query | Answer |
|---|---|
| 磁共振的禁忌症是什么? 
 What are the contraindications for MRI? | 装有心脏起博器,带铁磁性心血管支架,带铁磁性体内植入物和3月内孕妇慎检。 
 MRI should be used with caution in individuals with cardiac pacemakers, ferromagnetic cardiovascular stents, ferromagnetic implants, and pregnant women in their first trimester. |
| 经腹子宫附件彩超需要憋尿吗? 
 Do I need to hold my urine for transabdominal pelvic ultrasound? | 经腹子宫附件彩超需要憋急尿才能做检查。 
 Transabdominal pelvic ultrasound requires a full bladder for optimal examination. |

### A.2.6 Treatment Prediction

Examples of treatment prediction question-answer pairs are in Table 16.

### A.2.7 Medication Prediction

Examples of medicine prediction question-answer pairs are in Table 17.

### A.2.8 Drug QA

The final dataset comprises 137 question-answer pairs, with the distribution of intent categories as follows:

- Indications: 27.0%
- Adverse Reactions: 21.9%
- Dosage Instructions: 18.2%
- Traditional Chinese Medicine: 10.2%
- Special Populations: 8.0%
- Precautions: 6.5%
- Pharmacokinetics: 3.6%
- Contraindications: 3.0%
- Interactions: 1.6%

### A.2.9 Insurance QA

Two examples of the question-answer pairs in the IQA dataset are shown in Table 18.

### A.2.10 Mental Health QA

This task requires LLMs to generate appropriate responses based on the patient's concerns and detailed descriptions, addressing their issues and providing helpful advice or guidance. The dataset is derived from Yixinli, a Chinese mental health service platform with approximately 22 million users and over six hundred professional counselors. High-quality responses are provided by psychological counselors and volunteers. We filtered the dataset using keywords such as 'admission', 'hospitalization', 'post-surgery', and 'discharge'.

Table 14: Example of Doctor Patient Dialogue Summarization.

| Historical Dialogue | Response |
|---|---|
| 患者：宝宝打了流脑疫苗和百白破之后，发烧，烧退了之后，又咳嗽，请问需要治疗吗？是打疫苗引起的吗，嗓子有点发红。
医生：一般疫苗发热也就是一天左右，不会引起咳嗽嗓子红，估计宝宝感冒了。宝宝多大了？几号打的疫苗？
患者：13号打的，两岁了，如果这样是需要吃消炎药吗
医生：咳嗽次数多么？今天医生给看过了吗？
患者：晚上多，白天偶尔咳嗽，前天看的医生，只是嗓子有点红，没开消炎药
医生：普通感冒咳嗽，一般一周左右就好了。如果引起肺炎，可能要进一步检查，治疗一般1-2周。
医生：今天宝宝精神食欲如何？
患者：精神食欲都不错
医生：精神食欲好，偶尔咳嗽，可以观察看看，如果咳嗽有痰，可以口服祛痰药物，家里有药吗？可以口服三天祛痰药看看效果，如果无效，就再带宝宝去正规医院儿内科门诊看看，及时调整用药。
患者：有清热解毒的，没有祛痰的
医生：咳嗽有痰吗？晚上影响宝宝睡眠吗？
患者：感觉有痰，不影响睡眠
医生：如果有痰，可以口服氨溴索口服液祛痰。
患者：好的，谢谢
医生：观察宝宝精神食欲，咳嗽情况，没加重逐渐好转就继续吃药，一般一周左右就好了。如果无效就就医及时调整用药，可能需要查血常规C反应蛋白肺炎支原体等，听诊心肺，支气管炎就继续吃药，必要时口服消炎药了。
Patient: After receiving the meningococcal and DPT vaccines, the baby had a fever. After the fever subsides, the baby coughs again. Does he need treatment? Is the vaccine causing this? The throat is slightly inflamed. Doctor: A vaccine-related fever usually lasts about a day and does not cause a cough or throat inflammation. It's likely your baby has a cold. How old is your baby, and when was the vaccine administered?
Patient: The vaccine was given on the 13th. My baby is two years old. Should he take antibiotics?
Doctor: Is the cough frequent? Did a doctor examine your baby today?
Patient: The cough is worse at night and occasionally during the day. We saw a doctor the day before yesterday, who noted a slightly red throat but did not prescribe antibiotics. Doctor: A common cold with a cough usually resolves in about a week. If it leads to pneumonia, further examination and treatment might be necessary, typically lasting 1-2 weeks. How is your baby's mood and appetite today?
Patient: The mood and appetite are good.
Doctor: If his mood and appetite are good and his cough is occasional, you can observe him. If the cough has phlegm, you can give expectorant medicine. Do you have any medication? Try an expectorant for three days and see if it helps. If not, take your baby to a regular pediatric outpatient clinic for further evaluation and medication adjustment.
Patient: We have medicine for clearing heat and detoxifying, but not an expectorant.
Doctor: Is there phlegm in coughing? Does it affect the baby's sleep at night?
Patient: Coughing phlegm does not affect sleep.
Doctor: If there is phlegm, you can give Ambroxol oral solution as an expectorant.
Patient: Okay, thank you.
Doctor: Keep observing your baby's mood, appetite, and cough. If it doesn't worsen and gradually improves, continue the medication. It usually resolves in about a week. If there's no improvement, seek medical attention to adjust the medication. Blood tests, including a complete blood count, C-reactive protein, and Mycoplasma pneumonia, may be necessary, along with a chest examination. If it's bronchitis, continue the medication and use antibiotics if necessary. | 主诉：患儿发热、咳嗽。
现病史：患儿注射疫苗后出现发热，咳嗽，咳痰。

辅助检查：暂缺。
既往史：不详。
诊断：上呼吸道感染。
建议：口服祛痰止咳药物3天，若无好转及时去医院完善血常规、C反应蛋白等检查。

Chief Complaint: Baby with fever and cough.
History of Present Disease: After vaccination, the baby had a fever, cough, and phlegm. Auxiliary Examination: None available.
History of Past Disease: Unknown.
Diagnosis: Upper respiratory tract infection.
Advice: Administer expectorant and cough suppressant medication for three days. If there is no improvement, promptly visit a hospital for a complete blood count, C-reactive protein, and other tests. |

Table 15: Example question-answer pairs for disease prediction.

| Question | Answer |
|---|---|
| 女，45岁，无痛性肉眼血尿1个月，尿中偶有血块，伴膀胱刺激症状。B超见膀胱右侧壁有一个1cm×2cm软组织影，有蒂。应考虑的诊断是？可能的疾病包括：膀胱结石、急性膀胱炎、膀胱异物、膀胱憩室、膀胱肿瘤。 | 膀胱肿瘤 |
| Woman, 45 years old, has had painless gross hematuria for one month, occasionally with blood clots in the urine, accompanied by symptoms of bladder irritation. Ultrasound shows a 1cm x 2cm soft tissue shadow on the right wall of the bladder, with a pedicle. What should be the considered diagnosis? Possible diseases include: bladder stones, acute cystitis, foreign body in the bladder, bladder diverticulum, bladder tumor. | Bladder tumor |
| 女性，45岁。双手和膝关节肿痛伴晨僵1年。体检：肘部可及皮下结节，质硬，无触痛。诊断首先考虑？可能的疾病包括：系统性硬化症、骨关节炎、痛风、类风湿关节炎、风湿性关节炎。 | 类风湿关节炎 |
| Female, 45 years old. She has been experiencing swelling and pain in both hands and knee joints accompanied by morning stiffness for a year. Physical examination: subcutaneous nodules can be felt in the elbow area, which are hard and painless to touch. What should be the first consideration for diagnosis? Possible diseases include: systemic sclerosis, osteoarthritis, gout, rheumatoid arthritis, rheumatic arthritis. | Rheumatoid arthritis |

## A.3 Ethics

### A.3.1 Ethnics statement

Although this dataset is doctor-rewritten rather than directly sourced from patients, we have obtained ethical approval from the Institutional Review Board at Shanghai Children's Medical Center affiliated to Shanghai Jiao Tong University School of Medicine, Peking University People's Hospital and Xiangya Hospital (NO. SCMCIRB-K2022139-1, No.2019PHB163-01, No.202005120 respectively), which allowed the computational analysis of retrospectively acquired, de-identified text for research purposes. All datasets from the online platforms were obtained with the user's informed consent and used under specific data use agreements. All data were re-written and de-identified, ensuring that no personal information was disclosed.

Table 16: Example question-answer pairs for treatment prediction.

| Question | Answer |
|---|---|
| 经产妇，40岁。近2年痛经并逐渐加重，伴经量增多及经期延长，届时需服强止痛药。查子宫均匀增大如孕8周，质硬，有压痛，经期压痛明显。本例确诊后的处置应选择？可能的治疗方案包括：镇痛药物治疗、雌激素治疗、化学药物治疗、手术治疗、放射治疗。 | 手术治疗 |
| Multiparous woman, 40 years old. She has had dysmenorrhea for the past two years, which has gradually worsened, accompanied by increased menstrual flow and prolonged menstruation, requiring strong painkillers during this time. Examination shows that the uterus is uniformly enlarged to the size of an 8-week pregnancy, hard, with tenderness, and the tenderness is obvious during menstruation. What should be the choice of treatment after diagnosis? Possible treatments include: painkiller treatment, estrogen treatment, chemotherapy, surgical treatment, radiation therapy. | Surgical treatment |
| 女，47岁。从3米高处坠落，致左胸壁外伤3小时。查体：T：36.5℃，P120/分，R25/分，BP100/60mmHg，神志清楚，气管居中，胸壁反常呼吸，左胸壁可触及多根肋骨断端，左肺呼吸音明显减弱，最适宜的处理方法？可能的治疗方案包括：胸腔闭式引流、胸腔穿刺排气排液、开胸探查+肋骨固定、胸壁加压包扎、镇静止痛，肋骨固定。 | 镇静止痛，肋骨固定 |
| Female, 47 years old. She fell from a height of 3 meters, causing trauma to the left chest wall for 3 hours. Physical examination: T: 36.5℃, P120/min, R25/min, BP100/60mmHg, clear consciousness, trachea in the middle, abnormal chest wall breathing, multiple rib fractures can be felt on the left chest wall, significantly weakened breath sounds in the left lung. What is the most suitable treatment method? Possible treatment plans include: closed thoracic drainage, thoracic puncture for gas and fluid drainage, thoracotomy + rib fixation, chest wall compression bandage, sedation and pain relief, rib fixation. | Sedation and pain relief, rib fixation |

Table 17: Example question-answer pairs for medicine prediction.

| Question | Answer |
|---|---|
| 女，67岁。双下肢水肿1个月，既往高血压病史15年，未规范用药物治疗。查体：BP160/100mmHg，双下肢轻度凹陷性水肿，实验室检查：血肌酐97μmol/L，血钾3.4mmol/L，尿蛋白（++）。应首选的降压药物是？可能的药品包括：钙通道阻滞剂、a受体拮抗剂、噻嗪类利尿剂、血管紧张素转换酶抑制剂、β受体拮抗剂。 | 血管紧张素转换酶抑制剂 |
| Woman, 67 years old. She has had edema in both lower limbs for one month, with a history of hypertension for 15 years, without standardized medication treatment. Physical examination: BP 160/100mmHg, mild pitting edema in both lower limbs, laboratory tests: blood creatinine 97μmol/L, blood potassium 3.4mmol/L, urine protein (++). What should be the first choice of antihypertensive drug? Possible drugs include: calcium channel blockers, alpha receptor antagonists, thiazide diuretics, angiotensin-converting enzyme inhibitors, beta receptor antagonists. | Angiotensin-converting enzyme inhibitors |
| 男，50岁。吃海鲜后夜间突发左足第一跖趾关节剧烈疼痛1天。查体：关节局部红肿，压痛明显。既往无类似发作。化验：血尿酸602mmol/L。目前最主要的治疗药物是？可能的药品包括：苯溴马隆、别嘌醇、抗生素、非甾体抗炎药、甲氨蝶呤。 | 非甾体抗炎药 |
| Male, 50 years old. He suddenly experienced severe pain in the first metatarsophalangeal joint of his left foot after eating seafood at night. Physical examination: local joint redness and swelling, obvious tenderness. He has no history of similar episodes. Lab test: blood uric acid 602mmol/L. What is the main treatment drug currently? Possible drugs include: benzbromarone, allopurinol, antibiotics, non-steroidal anti-inflammatory drugs, methotrexate. | Non-steroidal anti-inflammatory drugs |

### A.3.2 Potential negative societal impacts

While the proposed benchmark encompasses 12 tasks across four stages of a patient's clinical journey, the scores obtained from this dataset should only be used as a reference. High scores achieved by LLMs on MedJourney do not necessarily imply their direct applicability in clinical settings. Validation from physicians is essential before any such implementation.

Table 18: Two question-answer examples in the IQA dataset.

| Question | Answer |
|---|---|
| Can teeth cleaning be covered by medical insurance? | The current teeth cleaning, periodontal disease treatment, tooth extraction, and filling most of the routine oral treatments are covered by medical insurance and can be paid with medical insurance. |
| Can diabetes be reimbursed by medical insurance? | Hello, diabetes is a chronic disease. It can be reimbursed by medical insurance. Generally, active medication for symptomatic treatment is needed. It can be managed by controlling diet and taking oral hypoglycemic agents or insulin injections. Additionally, regular follow-up checks of fasting blood sugar or urine routine tests are necessary. |

Table 19: Example of Mental Health QA.

| Historical Dialogue | Response |
|---|---|
| 问题：20岁女生，身体免疫功能紊乱，敏感多想该怎么办？

描述：身体免疫功能紊乱了，过敏，以前不会过敏（吃肉长荨麻疹），月经不调，从过敏开始，害怕自己身体有问题，以前从来不会的，左右胸部闷痛（去医院查过心脏，肺，血糖，甲状腺功能，都没有问题），怕冷，突然觉得自己身体冷，然后觉得自己好像得了重病。有过一次手术（小手术），然后觉得自己身体是有问题的，一点小问题容易多想，害怕死亡，怀疑自己有病，害怕给家人带来负担。这半年因为父母关系，有奔溃过，莫名其妙的哭，哭自己身体，哭家人。现在我自己认为我心里没有什么问题，就是容易多想，觉得自己身体有问题，不知道该怎么办。 | 您好，抱抱你，从你的字里行间让我感觉你应该年纪还比较小，你
现在担心的这些状态，身体上应该不排除有一些问题，具体可能去医院看看。在心理上，现在你可能是比较抑郁和悲伤的情绪。人的身和心态都是互相影响的，所以，我们也不能找出来说到底是谁先引起了谁，比说说是最近情绪的一些问题影响了身体，还是说是身体的一些毛病，让心情变得不好。没有必要去深究，建议您可以跟父母商量，找一个正规的医院去看一下，排除身体的一些疾病。心理上，你提到父母的关系不好，但父母之间是发生了怎样的事情，这个我觉得你可以说得更仔细一点，这样能帮你分析。如果真的是一些比较影响你的创伤性的事件的话，是需要做一些干预的。如果只是普通的比较爱多想这样的状态，就不用太管。当自己有一些比较负面的念头的时候，当你看到这些念头，就及时抽身出来，同时，让自己忙起来，按部就班的学习和工作，一切都会好起来，加油。 |
| Question: 20-year-old, female. I experienced immune dysfunction. Now, I am too sensitive to worry. What should I do? Description: I have developed immune dysfunction and allergies, which I did not have before (e.g., urticaria from eating meat). I also experience irregular menstruation since the onset of allergies and am anxious about my health, fearing serious illness. I have chest discomfort (heart, lungs, blood glucose, and thyroid function have been checked and are normal), cold intolerance, and a persistent feeling of being unwell. I had a minor surgery once, and since then, I feel like there is something wrong with my body. I tend to overthink minor issues, fear death, suspect that I am ill, and worry about burdening my family. In the past six months, I experienced emotional breakdowns due to my parents' conflicts, often crying inexplicably over my health and family. Currently, I think I have no psychological issues, but I tend to overthink and feel that there is something wrong with my body. I don't know what to do. | Hello, I understand your concerns. Given your age, it is possible that you are experiencing some physical health issues, which should be checked by a hospital visit. Psychologically, you may be dealing with depression and sadness. The mind and body affect each other, so it's difficult to determine whether emotional issues are causing physical symptoms or vice versa. There is no need to overanalyze this. I recommend discussing this with your parents and visiting a reputable hospital to rule out any physical illnesses. Regarding your psychological health, you mentioned your parents' relationship issues. If these issues are significantly impacting you, detailed communication can help you understand and address the problem. If traumatic events are affecting you, intervention might be necessary. However, if you are just prone to overthinking, it's less concerning. When you have negative thoughts, try to step back and keep yourself busy with routine studies and work. Everything will get better. Stay strong. |

## A.4 Limitations

The experiments were not conducted on more LLMs. This stems from the fact that some advanced LLMs, such as Med-PaLM, have not yet been made available.

