# OpenReview forum: "MedJourney: Benchmark and Evaluation of Large Language Models over Patient Clinical Journey"
_NeurIPS.cc/2024/Datasets_and_Benchmarks_Track — NeurIPS 2024 Track Datasets and Benchmarks Poster_

### Official Review · Reviewer_8tiK · 2024-07-04
**Nice piece of work on evaluating LLMs' performance on the medical domain**

**Rating:** 6
**Confidence:** 3

**Review:**

1. The authors underlined "the clinical process in patient care" is important and the dataset is constructed by following this concept. However, the authors did not explain anything on the introduction. Since the readers of NeurIPS may not expertise on the medical domain, the authors may want to elaborate what is "the clinical process" and why the concept is important at introduction. In the same time, the current manuscript

2. The description of the clinical process is inadequate, lacking justification for its relevance to LLM performance evaluation or its division into four steps. The authors reference footnotes for their explanation, but these sources are not reliable literature. Consequently, the authors may want to enhance the overall description of the clinical process and substantiate its importance. Additionally, it is essential to cite peer-reviewed papers or other credible sources to support their arguments.

3. Lack of explanation on the quality control. The dataset have been labeled by medical experts. However, I cannot find any mentioning on how the authors controlled the quality of the dataset. The authors should clarify this.

**Strengths:**

1) The authors constructed new data based on the concept of clinical process. The data has been labeled by medical experts, and the data can be an important contribution to the community.

2) The idea of ​​a dataset based on clinical concepts is very interesting, and I think it will be able to evaluate patient-centric LLM performance differently from existing data.

**Additional Feedback:**

N/A

**Clarity:**

The paper is easy to understand and explains their contribution appropriately.

1) However, if the authors explain the 12 tasks in the abstract simply, it would be helpful to potential readers to understand their work.

2) In line 55, the authors mentioned "To the best of our knowledge." This mention can be considered passing the responsibility. Although it have not been influenced my decision, but the authors may want to clarify this through additional literature review.

3) Line 64, LlamA -> Llama

4) References are missing for some models (e.g. Llama, QWen, ChatGLM, and Huatuo GPT).

5) In line 35-46, the authors may want to enhance the readability. The current version is a little bit difficult to read due to the presentation. I suggested to revise this part. For example,
"which can be categorized into three types: 1) Exam-based, 2) QA-based, 3) Task-based. {explanation for each type.}"

**Correctness:**

The authors said the data was labeled by medical experts (line 59). However, data quality control is not specified in the manuscript.

**Documentation:**

N/A

**Ethics:**

The authors said they built it based on actual patient data. However, according to the manuscript, the authors said they would make the data publicly available on GitHub (supplement line 227). I am asking AC and the authors to confirm whether this is not a violation of privacy. Note that I am willing to revise the ethics flag and adjust my score as soon as it is confirmed that there is no privacy violation. (Solved)

**Limitations:**

The authors reported the limitation of their work well. In addition, the authors discussed the potential negative social impact. I did not find any issue on this section.

**Opportunities For Improvement:**

Please check the review.

**Relation To Prior Work:**

The authors introduced previous works and the difference of their work well.

**Summary And Contributions:**

This paper introduce a new benchmark dataset to assess LLMs' performance on the medical domain inspired by the clinical process. The dataset consists of 12 tasks which 5 stem from previous dataset and the authors construct 7 new tasks (DR, PCD, HQA, IQA and DQA). The experiments conducted on diverse open and closed LLMs and the authors reported baseline performances. Overall, the idea to use clinical process is interesting and the contributions are enough to be published. However, not only is there still a lack of evidence and explanation for key concepts, but there are also questions about data quality.

---

> ### Author Rebuttal · Authors · 2024-08-17
>
> Thanks for your helpful comments, below are the replies to your questions. In the following, R
>  denotes review, L denotes Limitation,  CO denotes Correctness and CL denotes Clarity. And AR, AL, ACO, ACL denote the answers to R, O, L, CO and CL respectively.
>
>
> **R1:** clinical process in patient care concept
>
> **AR1:** In this paper, we segment the clinical process based on the patient's journey, specifically focusing on the stages before visiting the hospital (planning), during the hospital visit (access and delivery), and after the hospital visit (ongoing care).
>
> During the planning stage, patients experiencing symptoms are guided to the appropriate department through Department Recommendation (DR). The Pre-Consultation Dialogue (PCD) collects the patient's main complaints in a dialog format and summarizes them for the doctor's review. The Hospital Reception QA (HQA) informs patients about the specifics of their hospital visit, such as what items to bring, dietary restrictions before certain examinations, and so on.
>
> In the access stage, the Doctor Response Generation (DRG) provides potential responses to the doctor based on the patient-doctor conversation. The Patient Doctor Dialogue Summarization (PDDS) further condenses the dialogue for easy reference.
>
> During the delivery stage, Examination Prediction (EP), Disease Prediction (DP), Treatment Prediction (TP), and Medication Prediction (MP) recommend potential actions for the doctor's reference, such as suggesting further examinations, possible diagnoses, appropriate treatments, and medications for the patient.
>
> In the ongoing care stage, Drug QA (DQA) instructs patients on medication usage and related knowledge. Insurance QA (IQA) provides information on medical insurance, and Mental Health QA (MQA) focuses on the patient's mental well-being.
>
> While the proposed 12 tasks do not cover all aspects of a clinical process, such as radiology report generation, they effectively connect the dots of a patient's journey.
>
>
>
> **R2:**  lacking justification for its relevance to LLM performance evaluation or its division into four steps
>
> **A2:** In this paper, we propose a division of the patient's clinical journey into four stages: planning, access, delivery, and ongoing care. This segmentation aligns with similar divisions proposed in other studies, beyond the ones referenced in the footnotes.
>
> For instance, in [1], the authors segment the patient journey into six stages: 1) Initial Contact and Symptom Recognition; 2) Diagnostic Evaluation and Treatment Planning; 3) Treatment and Care Delivery; 4) Follow-up Care and Monitoring; 5) Long-term Management and Disease Prevention; 6) Patient Education and Empowerment. Here, stage 1 aligns with our "planning" stage, stage 2 with "access", stage 3 with "delivery", and stages 4 to 6 with "ongoing care".
>
> In [2], the authors define the patient journey as "encounters with healthcare facilities, a hospital unit, a specialist visit, a primary care clinic, a home health agency". In this definition, "healthcare facilities" corresponds to our planning stage, "hospital unit" and "specialist visit" align with the access and delivery stages, and "primary care clinic" and "home health agency" map to the ongoing care stage.
>
> In [3], the authors describe a patient's journey through surgery, which includes seven stages: 1) Out-patient visit, 2) Examination at out-patient clinics, 3) Hospitalization and surgery, 4) Post-surgical care, 5) Discharge, 6) Rehabilitation Stay, 7) Follow-up visit. Here, stage 1 corresponds to "planning", stage 2 to "access", stages 3 and 4 to "delivery", and stages 5 to 7 to "ongoing care".
>
> [1] Sehrawat, Sunil Kumar. "Empowering the Patient Journey: The Role of Generative AI in Healthcare." International Journal of Sustainable Development Through AI, ML and IoT 2.2 (2023): 1-18.
>
> [2] Donaldson, Liam, et al. "Textbook of patient safety and clinical risk management." (2021): 496.
>
> [3] Gualandi, Raffaella, et al. "Exploring the hospital patient journey: what does the patient experience?." PloS one 14.12 (2019): e0224899.
>
> We will add these references in the revision of this paper.
>
> **R3:** Quality control
>
> **AR3:** In this paper, we present 12 tasks, seven of which are built from existing corpora. These have been extensively used in current research, ensuring their quality. The five newly proposed datasets, namely Department Recommendation (DR), Pre-Consultation Dialogue (PCD), and Hospital Reception QA (HQA), Drug QA (DQA) and Insurance QA (IQA), originate from Tencent Health Solution. These datasets have been annotated by professional doctors, ensuring their relevance and accuracy.
>
> In addition to the authors who are affiliated with hospitals, we also have a strong medical editorial team (https://baike.qq.com/), which further enhances the quality of our work. For the three QA tasks, we have implemented a rigorous review and modification process to ensure the highest level of accuracy and relevance; For the Department Recommendation (DR) and Pre-Consultation Dialogue (PCD), we employ experienced physician to recompose the data sets based on patients' inputted queries to ensure the medical correctness.
>
> We will provide additional relevant information in the appendix, including the specific data quality control process.

---

> > ### Author Rebuttal · Authors · 2024-08-17
> >
> > **CL1:** Explain the 12 tasks in the abstract simply
> >
> > **ACL1:** We will add more descriptions of the 12 tasks in the abstarct
> >
> > **CL2:**  additional literature review
> >
> > **ACL2:** Thank you for your suggestion. We will revise the phrase "to the best of our knowledge." We have summarized the literature on medical evaluation and presented it in Table 1, which includes recent benchmarks in Chinese. We will also reference the newly published paper of MedBench [1]. There are numerous English-based medical benchmarks, and we have summarized some works in the related works section and will add more literature review in next revision. We will also cite and discuss [2-4] provided by reviewer G1cc.
> >
> >
> > [1] M, Hu W, Ding J, et al. "MedBench: A Comprehensive, Standardized, and Reliable Benchmarking System for Evaluating Chinese Medical Large Language Models." Big Data Mining and Analytics, 2024, https://doi.org/10.26599/BDMA.2024.9020044
> >
> > [2] AgentClinic: a multimodal agent benchmark to evaluate AI in simulated clinical environments
> >
> > [3] NoteChat: a Dataset of Synthetic Doctor-Patient Conversations Conditioned on Clinical Notes
> >
> > [4] DR.BENCH: Diagnostic Reasoning Benchmark for Clinical Natural Language Processing
> >
> >
> > **CL3 & CL4** type and references to LLMs
> >
> > **ACL3 & ACL4:**  We will fix the typos and add reference of these LLMs.
> >
> > **CL5:** Explain the three types 1) Exam-based, 2) QA-based, 3) Task-based
> >
> > **ACL5:** For the Exam-based tasks, questions are typically selected from the United States Medical Licensing Examination (USMLE) or the China National Medical Licensing Examination (CNMLE). These questions usually come in the form of a problem, multiple-choice options, and an answer.
> >
> > For instance:
> >
> > > Question:
> > A 77-year-old male presents with progressive right-hand tremors and slow movements. The patient has a history of benign prostatic hyperplasia and mild renal insufficiency. Which medication would be most appropriate for his treatment?
> > > Candidate Options:
> > (A) Artane (B) Levodopa (C) Selegiline (D) Amantadine (E) Bromocriptine
> > > Answer:
> > (B) Levodopa
> >
> > For the QA-based tasks, an example can be found in Table 5.
> >
> > > Question: What are the side effects of Carbamazepine?
> > Answer: Common adverse reactions to Carbamazepine primarily include dizziness, drowsiness, fatigue, nausea, and other gastrointestinal symptoms. It can also cause bone marrow suppression, toxic hepatitis, and, more rarely, exfoliative dermatitis.
> >
> > For Task-based tasks, one example involves extracting named entities from plain text.
> >
> > > Prompt: Please identify the medical name entity in this sentence, "Tetanus Spasm toxin has a long-term effect on the autonomic nerve."
> > Target: Autonomic nerve; Tetanus spasm toxin
> >
> >
> >
> > **E1:** Ethic and privary
> >
> > **EA1:**
> > Thanks for your insightful comment!
> > In this paper, we present 12 datasets, seven of which are constructed from existing public datasets.
> >
> > Regarding the newly proposed datasets, Hospital Reception QA (HQA), Drug QA (DQA), and Insurance QA (IQA) primarily focus on medical knowledge and do not contain patients' personal information. For instance, as demonstrated in Table 5 in the paper, and Tables 2 and 7 in the supplementary documents, the questions pertain to drug usage, hospital visitation knowledge, and insurance. These datasets contain minimal personal information, a fact we have confirmed with human experts.
> >
> > The Department Recommendation (DR) and Pre-Consultation Dialogue (PCD) datasets are derived from users' inquiries prior to doctor visits. Therefore, these datasets focus on patients' primary complaints and do not include examination, diagnosis, or medication information.
> >
> > For the Department Recommendation (DR) and Pre-Consultation Dialogue (PCD) datasets, we also employ experienced physicians to recompose the data instead of directly using patients' input, further ensuring the absence of patient information. For example,
> >
> > |orignial patients' data	|rewritten patients' data|
> > |-----------|-------|
> > |My complexion is yellowish, and I have low physical strength. I easily get short of breath and frequently feel nauseous.|	My skin appears yellowish, and I experience an accelerated heartbeat and shortness of breath even with slight exertion. I constantly feel a bit nauseous. Which medical specialty should I consult?|
> > |My mother is 70 years old. She started experiencing sudden pain in her knees 10 days ago, and it has persisted without improvement even after resting for several days. In fact, it has worsened. Which medical specialty should she consult?	|Elderly person with sudden knee joint pain and limited mobility. Symptoms have worsened over the past 10 days.|
> >
> > Even though we did not directly use patients' data, we still obtained ethical approval from the hospitals.

---

> > > ### Comment · Reviewer_8tiK · 2024-08-19
> > > **Reviewer's comment on the data privacy**
> > >
> > > I do not know much about patient data protection in China. Generally, in the EU and US, a legal license and data use agreement are required to use even data that has been altered from actual patient data. This is a separate issue from de-identification. Can the authors present whether there are any legal issues with authors opening their data to the public, and if a license is required, how will this be managed? My concerns would be completely resolved if the ethics statement included related laws.

---

> > > > ### Author Rebuttal · Authors · 2024-08-24
> > > >
> > > > Among the five newly proposed datasets, Hospital Reception QA (HQA), Drug QA (DQA), and Insurance QA (IQA) are primarily composed of medical knowledge, containing minimal patient-specific information.
> > > >
> > > > Department Recommendation (DR) incorporates patients' self-reported symptoms, while Pre-Consultation Dialogue (PCD) includes patients' interactions with a chatbot. As DR and PCD are facilitated through a mobile app before doctor consultations, this data does not form part of traditional hospital records. Instead, they resemble health-related inquiries commonly found in search engines.
> > > >
> > > > We did not directly employ the DR and PCD data. Rather, we transformed the data to ensure privacy, a process that eliminates any personal information. The revised content significantly deviates from the patients' original entries, as demonstrated in the examples provided in response to your previous questions.
> > > >
> > > > As the reviewer correctly noted, China, like the EU and the US, mandates that medical data be anonymized prior to release. As per the Healthcare Data Security Guidelines set by the Standardization Administration of China (http://www.phic.org.cn/zcyjybzpj/bzypj/bzgf/gjbz/202103/P020210331605989883649.pdf, Page 6), the data, once anonymized, can be utilized and disseminated for purposes such as scientific research, medical/health education, public health, or healthcare operations.
> > > >
> > > > In addition to adhering to the aforementioned regulation, we also meticulously reviewed related work on medical datasets [1-9] and thoroughly assessed potential privacy and safety risks before undertaking a systematic process to curate our data. This process includes IRB approval, de-identification, rewriting, and human verification, as outlined in our Supplement. Ethics. We will limit the use of the MedJourney benchmark to academic research only.
> > > >
> > > >
> > > > [1] Liu, Wenge, et al. "Meddg: A large-scale medical consultation dataset for building medical dialogue system." (2020). The authors released patient-doctor conversation data sourced from Doctor Chunyu (https://www.chunyuyisheng.com/), an online Chinese medical consultation website.
> > > >
> > > > [2] Chen, Wei, et al. "A benchmark for automatic medical consultation system: frameworks, tasks and datasets." Bioinformatics 39.1 (2023): btac817. The authors released patient-doctor conversation data from Muzhi (http://muzhi.baidu.com), a Chinese online health community that provides professional medical consulting services for patients.
> > > >
> > > > [3] Sun, Hao, et al. "PsyQA: A Chinese Dataset for Generating Long Counseling Text for Mental Health Support." ACL-2021. The authors released mental health inquiries from Yixinli (https://www.xinli001.com/qa), a Chinese online mental health service platform.
> > > >
> > > >
> > > > [4] Zhang, Yuanzhe, et al. "MIE: A medical information extractor towards medical dialogues." Proceedings of the 58th Annual Meeting of the Association for Computational Linguistics. 2020. The authors released patient-doctor conversation data, also from Doctor Chunyu.
> > > >
> > > > [5] Zhang, Ningyu, et al. "Cblue: A chinese biomedical language understanding evaluation benchmark." arXiv preprint arXiv:2106.08087 (2021). In the CBLUE benchmark, the task CHIP-CDEE consists of 2,485 medical records, the task KUARK-IR consists of 1 million health related queries.
> > > >
> > > > [6] Xu Lin et al. "End-to-End Knowledge-Routed Relational Dialogue System for Automatic Diagnosis." AAAI-2019. The authors release the medical dialogues from Dingxiang Doctor(https://dxy.com/) and their annotated medical entities.
> > > >
> > > >
> > > > [7] Li, Dongdong et al. "Semi-Supervised Variational Reasoning for Medical Dialogue Generation." SIGIR-2022. The authors release the large-scale medical consulation datasets involved multi department and clinical scenarios.
> > > >
> > > >
> > > > [8] Zeng, Guangtao et al."MedDialog: Large-scale Medical Dialogue Datasets." EMNLP-2020. The authors release 3.4 million Chinese dialogues from Haodaifu (https://www.haodf.com/) and 0.26 million English consultation from iclinic (https://www.icliniq.com/) and healthcaremagic (https://www.healthcaremagic.com/)
> > > >
> > > >
> > > > [9] Kilicoglu, Halil, et al. "Semantic annotation of consumer health questions." BMC bioinformatics 19 (2018): 1-28. The researchers of National medical library (US) release the inquiry email from patients and the corresponding response from medical experts

---

> > > > > ### Comment · Reviewer_8tiK · 2024-08-27
> > > > > **Reviewer's response on the authors' comment**
> > > > >
> > > > > My ethical concerns has been released. I will change the ethical flag statement. Anyway, I really appreciate the authors on their hard work to relief my concerns.

---

> > > > > > ### Author Response · Authors · 2024-08-28
> > > > > >
> > > > > > We sincerely appreciate your insightful comments. Addressing these concerns is crucial for enhancing the quality of our paper. It has been our privilege to engage in discussions with you.

---

> > ### Comment · Reviewer_8tiK · 2024-08-19
> > **Reviewer's reply on the quality control**
> >
> > I understand that the authors have worked hard for quality control. However, even though the data have been made up of medical experts disagreements between data annotators may still exist. Can the authors disclose the list of annotators in the revised paper? It is okay if your real name is not mentioned. In fact, many clinical NLP papers provide information about annotators to increase trust in them [1]. Also, if possible, please also present the agreement scores of the annotators. If you have already provided an agreement score in the manuscript and I have missed it, please let me know.
> >
> > [1] https://aclanthology.org/2024.naacl-long.244.pdf

---

> > > ### Author Rebuttal · Authors · 2024-08-24
> > >
> > > Below is the list of annotators (we have contracts with these doctors):
> > >
> > > 1. Physician, Dr. Wu, Guangzhou First People's Hospital
> > > 2. Physician, Dr. Wang, Zhengzhou First People's Hospital
> > > 3. Physician, Dr. Li, Third Affiliated Hospital of Southern Medical University
> > > 4. Physician, Dr. Hu, First Affiliated Hospital of Hunan University of Chinese Medicine
> > > 5. Physician, Dr. Ma, Beijing Lejian Dongwai Outpatient Department
> > > 6. Physician, Dr. Yang, Shanghai Xinshijie Eye Hospital
> > > 7. Attending Physician, Dr. Peng, Tongling People's Hospital
> > > 8. Attending Physician, Dr. Li, Taikang Tongji (Wuhan) Hospital
> > > 9. Attending Physician, Dr. Ren, Wuhan Asia Heart Hospital
> > >
> > >
> > > Regarding the Department Recommendation (DR) dataset, we possess the 'gold standard' labels since we are aware of the department the patient visits, and there is no negative feedback. Consequently, the doctor's task is to select and rewrite the patients' descriptions of their symptoms. As detailed in the supplementary material,
> > >
> > > ```
> > > In the first step, each department was initially assigned to a doctor who composed more than five patient complaints specific to that department. In the second step, another doctor supplemented potential departments based on the patient complaints generated in the first step. Finally, the meta
> > > annotator evaluated each case, retaining the top five cases of the highest quality for each department.
> > > ```
> > > Given that the process doesn't involve voting or categorization, there is few disagreement between the annotator and the meta-annotator. They collaborate effectively to accomplish this task.
> > >
> > > In the case of the Pre-Consultation Dialogue (PCD) dataset, doctors have selected 100 representative conversations and rewritten them to eliminate any personal information. The summarized content is derived directly from these dialogues. Once again, there is few disagreement between the annotator and the meta-annotator.
> > >
> > > Regarding the Hospital Reception QA (HQA), Drug QA (DQA), and Insurance QA (IQA) datasets, the questions and answers are grounded in medical knowledge rather than patient-specific data. The responses are provided by professional doctors, leveraging their expertise. This information embodies common medical knowledge, which is generally accepted without dispute.

---

> ### Comment · Reviewer_8tiK · 2024-08-19
> **Anyway**
>
> Anyway other explanations on my questions handled well. I adjusted my score.

---

### Official Review · Reviewer_G1cc · 2024-07-08

**Rating:** 5
**Confidence:** 4

**Review:**

Pros:

1. High impact: This work is a significant advancement in the field of medical AI. And focuses on real-world clinical questions rather than static medical question-answering.

2. Comprehensive Coverage: This work segments the patient's clinical journey into planning, access, delivery, and ongoing care.
and introduces 12 datasets for different tasks associated with these stages.

Cons (see Opportunities For Improvement for more detail):

1. Missing human evaluation or metrics based on LLM.

2. Limited discussion on potential biases.

3. Authors only described the result, but did not answer the key question proposed in the introduction (line 051): whether LLMs could assist patients in real clinics. For example, Is B-4 of 21.316 high enough to assist patients in real clinics? If not, it also misses an interpretation of how current LLMs can be integrated into clinical workflows. [1]

4. Missing Chinese Private LLMs as baselines.

[1] https://www.nature.com/articles/s41746-024-01083-y

**Strengths:**

The introduction of the MedJourney benchmark is a significant advancement in the field of medical AI. Previous ClinicalNLP research benchmarks focus on text-book multiple-choice questions from exams or real-world information extraction tasks such as named entity recognition. These benchmarks do not measure the ability of LLM to solve real-world clinical questions, which is the aim of this research.

**Additional Feedback:**

NA

**Clarity:**

(1) The prediction in “Disease Prediction” is likely to mislead readers. Prediction typically predicts patient outcome in the future say 6 months given current status [1], but “Disease Prediction” here means Disease Diagnosis at the current timepoint.

(2) It is best to tell readers what is in the supplementary material in line 249. For instance, an example of the generated Doctor Patient Dialogue Summarization is shown in the supplementary material table xxx.

[1] Laila Rasmy. Med-BERT: pretrained contextualized embeddings on large-scale structured electronic health records for disease prediction

**Correctness:**

(1) In line 247, the author said that they used NLG metrics, such as BLEU, ROUGE, and entity-based metrics, but in table 6, the result was only on B-4, which is short for BLEU-4. Where are the results for ROUGE?

(2) The automatic benchmark for NLG task is based on bleu, which is too simple and may not align with human pyhysician judgment.

(3) PCD dataset is called Pre-Consultation Dialogue in line 94, it utilizes a multi-turn conversation to gather patient info. However, in method section line 126, it is implemented as a conversation summarization task without any intent of information gathering. The authors are overselling their story.

**Documentation:**

Missing info on how to test a new LLM on this benchmark.

**Ethics:**

No ethical issue.

This submission involves human subjects, but it is IRB approved. The authors de-identified text, ensuring that no personal information was disclosed.

**Limitations:**

While the MedJourney benchmark is comprehensive, its focus on Chinese datasets limits its applicability to non-Chinese contexts. Including multilingual datasets or benchmarks applicable to various healthcare systems globally would enhance its significance and broader impact.

The authors could also discuss the long-term societal impacts of deploying LLMs in healthcare. This could include considerations of how these technologies might change the roles of healthcare professionals [1], the patient education[2], and the accessibility of healthcare services [3].

[1] AJ Thirunavukarasu. Large language models will not replace healthcare professionals: curbing popular fears and hype

[2] P Cai. PaniniQA: Enhancing Patient Education Through Interactive Question Answering

[3] X Wang ChatGPT: promise and challenges for deployment in low- and middle-income countries

**Opportunities For Improvement:**

(1) The study relies heavily on automated metrics for evaluation (such as BLEU), which might not capture the nuanced performance aspects of LLMs in clinical settings [1]. Incorporating qualitative assessments by clinical experts could provide deeper insights into the practical utility of the models.

(2) Although the paper addresses ethical considerations, it could benefit from a more detailed discussion on the potential biases in the datasets and the models. Analyzing and mitigating biases related to gender, age, socioeconomic status, and other factors would strengthen the ethical framework of the research [2]. The author should include this information for each question in the access and delivery.

(3) While the benchmark is designed to assess LLM performance in clinical settings, the study could benefit from an exploration of how these models can be integrated into clinical workflows. Practical guidelines or case studies demonstrating successful implementations of LLMs in healthcare could provide valuable insights for practitioners.

(4) Lack of Chinese Private LLMs as baseline: Since English Private LLMs (ChatGPT and GPT-4) perform so bad in most tasks as shown in Table 7. could Chinese Private LLMs such as Wenxinyiyan further improve performance on a benchmark in Chinese?

(5) Lack of evaluation on patient information gathering. The authors specified the importance of Planning for the patient, however, it does not evaluate how LLM can gather info to plan for pateint.

[1] Asma Ben Abacha. An Investigation of Evaluation Metrics for Automated Medical Note Generation

[2] O Shaikh. On second thought, let’s not think step by step! Bias and toxicity in zero-shot reasoning.

**Relation To Prior Work:**

This paper misses some references to previous works.

AgentClinic is adapted from Objective Structured Clinical Examination (OSCE) [1]. It is designed to test performance in a simulated clinical setting and competence in skills such as communication, clinical examination, medical procedures, and time management. This overlaps with the 4 stages proposed in this work.

NoteChat is a Dataset of Synthetic Doctor-Patient Conversations Conditioned on Clinical Notes [2]. The question answering between doctors and patient is similar to 2nd stage proposed in this work.

DR.BENCH is a diagnostic reasoning benchmark for clinical natural language tasks [3]. While DR.BENCH missed the conversation task proposed in this submission, DR.BENCH has multiple NLP overlaps with this work, including summarization, clinical note question answering, and delivery prediction.



[1] AgentClinic: a multimodal agent benchmark to evaluate AI in simulated clinical environments

[2] NoteChat: a Dataset of Synthetic Doctor-Patient Conversations Conditioned on Clinical Notes

[3] DR.BENCH: Diagnostic Reasoning Benchmark for Clinical Natural Language Processing

**Summary And Contributions:**

This study focuses on the application of large language models (LLMs) in the medical domain. The study introduces a new benchmark dataset, MedJourney, which is designed to evaluate LLMs' performance across the entire clinical journey of a patient. The MedJourney consists of 12 tasks across 4 stages: planning, access, delivery, and ongoing care. In addition, they evaluate the performance of existing LLMs on MedJourney. The authors also claim that this is the first work that organizes benchmark w.r.t patient clinical journey. (But this more not be true, see Relation to prior work for more detail)

---

> ### Author Rebuttal · Authors · 2024-08-17
>
> Thanks for your helpful comments, below are the replies to your questions. In the following O denotes oppotunity, L denotes Limitation,  CO denotes Correctness and CL denotes Clarity. And AO, AL, ACO, ACL denote the answers to O, L, CO and CL respectively.
>
> **O1:** Missing human evaluation or metrics based on LLM
>
> **AO1:** Thansk for your constructive suggestion. For the proposed benchmark, we would like the evaluation process conducted automatically, which can be used to guide the building of large language models. Therefore in the paper, we only conduct the automatic evaluation of the LLMs on the proposed Medjourney benchmark. In addition to the NLG metrics like BLEU and the accuracy metrics, we also conduct entity based metrics in Table 8 which can evaluate whether the core information is obtained.
>
> Furthermore, we also leverage the LLMs to evaluate the performance of LLMs. We let GPT-4 to rate the performance of LLMs on benchmark, espeically for the QA questions. If we average the performance on these 7 tasks. QWen 72b, GPT4 and QWen 32b rank the top 3.
>
>
> | Model     | PCD   | HQA   | DRG   | PDDS  | DQA   | IQA   | MQA |
> |-----------|-------|-------|-------|-------|-------|-------|-------|
> | chatglm3  | 80.16 | 78.34 | 81.83 | 82.35 | 82.66 | 82.41 | 87.96 |
> | qwen7b    | 87.37 | 83.32 | 85.83 | 86.67 | 87.32 | 87.53 |  90.3 |
> | qwen14b   | 89.37 | 85.58 | 81.69 | 87.06 | 87.68 | 86.09 | 89.44 |
> | qwen32b   | 88.24 |  85.8 | 86.22 | 87.32 | 88.13 | 87.53 | 90.08 |
> | qwen72b   | 89.34 | 86.57 | 85.64 | 87.84 | 88.88 | 86.42 | 89.45 |
> | chatgpt   | 88.18 |  83.1 | 69.95 | 86.05 | 83.43 | 84.06 | 89.89 |
> | gpt4      | 89.44 | 85.86 | 85.86 | 87.51 | 87.80 | 87.33 | 90.19 |
> | huatuo7b  | 81.89 | 79.41 | 82.94 | 81.41 | 87.24 | 84.99 | 88.61 |
> | huatuo34b | 89.11 | 82.44 | 84.58 | 85.75 | 86.34 | 86.28 | 89.38 |
> | medllm    | 72.45 | 78.89 | 81.83 | 77.69 | 79.37 | 77.01 | 86.31 |
>
>
>
>
> We will cite your provided paper, we totally agree that LLM only serves as an assitant and is not able to replace doctors.
>
> **O2:** The author should include this information for each question in the access and delivery.
>
>
> **AO2:** We conduct the bias evaluation.
>
>
> **O3:** whether LLMs could assist patients in real clinics.
> **AO3:** Many proposed tasks have been already deployed in clinics. In the Tencent Smart Hospital Solution (https://healthcare.tencent.com/en), the Department Recommendation (DR), Hospital Reception QA (HQA) and Pre-Consultation Dialogue (PCD) have been deployed to more than 1500 hospitals in China (more than 600 are top-lier hospitals), with more than 1.5 million month active users. These three tasks are relating to patients information understanding and gathering, before visiting the doctors. All these three tasks are initially built with deep learning approaches but now upgraded to LLM approaches.
>
> For Doctor Response Generation (DRG) and 4 tasks in the access stage, they have been deployed as a part of HIS (Health Informaiton System), since they are locally built into hospitals, we are not aware of the actual usage. But all these applications are used for doctors' reference, the decisions are made by doctors.
>
>
> **O4:** Missing Chinese Private LLMs as baselines
>
> **AO4:** Thanks for your great advices. We have conducted experiments on one popular Chinese LLM (Tencent Hunyuan https://hunyuan.tencent.com/), we try on two versions of Hunyuan, the standard version and the advanced version. We can find that the private Chinse LLM (advanced version Hunyuan) ranks 1st on 9 of 12 tasks and signiciantly outperform GPT-4 on 10 tasks.
>
>
> | Model | DR Acc| PCD B-4|HQA B-4|PDDS B-4|DRG B-4|	EP Acc|	DP Acc|	TP Acc|	MP Acc|	DQA B-4|IQA B-4|MQA B-4|AVG Acc|AVG B-4|
> |:------:|:------:|:------:|:------:|:------:|:-----:|:-----:|:-----:|:-----:|:-----:|:-----:|:-----:|:-----:|:-----:|:-----:|
> |hy standard zero-shot|0.308|11.035|1.960|12.638|2.647|0.597|0.529|0.453|0.615|8.633|3.543|4.053|
> |hy standard one-shot|0.398|16.868|4.337|21.342|3.681|0.433|0.430|0.331|0.402|7.399|5.319|4.043|
> |hy advanced zero-shot|0.316|8.309|2.210|5.333|2.209|0.771|0.752|0.635|0.831|8.351|3.806|4.101|
> |hy advanced one-shot|0.424|21.846|7.585|17.228|3.208|0.755|0.804|0.750|0.790|8.279|5.121|3.944|
>
>
> **O5:** Lack of evaluation on patient information gathering
>
> **AO5:** The planning stage includes Department Recommendation (DR), Pre-Consultation Dialogue (PCD) which are patient specific. While Hospital Reception QA (HQA) includes general instruction knowledge that are common to all patietns. For DR and PCD, they are conducted prior to doctor visits. Currently, the tasks are done by patients via mobile APP of hospitals. Since the information are inputted by patients, they could be erroreous. Therefore, we also employ experienced physicians to recompose the data instead of directly using patients' input, In addition to the authors who are affiliated with hospitals, we also have a strong medical editorial team (https://baike.qq.com/), which further enhances the quality of our work.

---

> > ### Author Rebuttal · Authors · 2024-08-17
> >
> > **L1:** The authors could also discuss the long-term societal impacts of deploying LLMs in healthcare.
> >
> > **AL1:** Thanks for pointing out these three related works, we will cite and reference these works in next revision.
> >
> > For the long-term societal impact, we will take the LLM enabled Pre-Consultation Dialogue (PCD) deployed in Shenzhen Renmin Hospital for example, till now, it has served 100 thousands patients. On the doctor side, the PCD achieves a 75% reduction (p-value<0.001) in the total time of writting notes for doctors. As a results, the doctors have more time to meet patients. On the patient sides, it achieves an increase of 28% in patients' saficatory degeee.
> >
> >
> > **CO1:** The metrics for ROUGE
> >
> > **ACO1:** Thanks for your careful reading! Below are the ROUGE performance and we included them in our local version,
> >
> >
> > | Model     | PCD   | HQA   | DRG   | PDDS  | DQA   | IQA   | MQA |
> > |-----------|-------|-------|-------|-------|-------|-------|-------|
> > |chatglm3-zeroshot-output|21.471|9.968|11.081|20.746|18.107|16.476|17.736|
> > |qwen7b-zeroshot-output|14.424|7.592|10.264|21.176|16.93|14.317|16.228|
> > |qwen14b-zeroshot-output|20.336|8.831|14.815|21.155|18.437|15.717|17.021|
> > |qwen32b-zeroshot-output|18.47|7.881|12.114|19.298|18.929|14.67|17.343|
> > |qwen72b-zeroshot-output|21.143|8.796|12.206|23.981|18.851|15.7|16.958|
> > |chatgpt-zeroshot-output|23.639|10.47|13.671|25.796|18.53|17.431|16.999|
> > |gpt4-zeroshot-output|14.461|6.29|8.878|20.907|16.435|13.413|15.407|
> > |huatuo7b-zeroshot-output|12.621|8.869|8.255|20.006|18.464|14.883|17.383|
> > |huatuo34b-zeroshot-output|19.069|7.923|10.811|26.995|18.779|14.942|17.356|
> > |medllm-zeroshot-output|10.998|9.785|7.801|16.364|17.41|15.602|15.61|
> >
> >
> >
> > **CO2:** Overselling the story of PCD
> >
> > **ACO2:** The Pre-Consultation Dialogue is the multi-turn dialog between patient and chatbot. The LLM generated the summary of patients and chatbot, and present it to the doctor before actually visiting the patients. In this manner, the doctor can easily copy it to the notes for more effective face to face diagnosis. Currently PCD is not purely LLM based solution but a hybrid system of pred-defined template and LLMs. For the next reply prediction, Doctor Response Generation (DRG) predicts the next reply for doctors given historical patient and doctor dialog.
> >
> > **CL1:** Disease Prediction clarication.
> >
> > **ACl1:** Your understanding is totally correct, it means Disease Diagnosis at the current timepoint. We will replace certain terms with phrases like "disease diagnosis" to avoid confusing readers.
> >
> >
> > **CL2:**  tell readers what is in the supplementary material in line 249.
> >
> > **ACL2:** Thanks for your helpful advices! We will add the reference to the figure and tables supplementary material in the main paper.
> >
> >
> >
> > **R1:** This paper misses some references to previous works
> >
> > **AR1:** Thanks for providing these three related works, we will cite and discuss these three papers in the Table 1. The work you mentioned is indeed related to ours, but it does not overlap.

---

> > ### Comment · Reviewer_G1cc · 2024-08-17
> > **Follow up**
> >
> > Thank author for their detailed response. However, I still have the following questions/comments:
> >
> > Question 1: Regarding AO1,
> > I find the evaluation metrics used in the study to be somewhat simplistic, especially considering that metrics are a crucial aspect of a benchmark. Do the B-4 scores or GPT-4 ratings align with human preferences? If I were to develop a language model that outperforms yours based on B-4 scores, would you or your hospital adapt to my model? Additionally, could the authors provide more details about the methodology behind the GPT-4 ratings? For example, how does GPT-4 determine which of the two outputs is superior?
> >
> > Question 2: Regarding AO2, B-4 score of about 4.0 or entity recall of about 30% are relatively low scores. This result represents that LM are not in agreement with doctor generated references. Could author provide case study to find disagreements/issues of current best model? This would pave the road for future works in this direction.
> >
> > Comment 1: Regarding ACO2,
> > Thank the author for confirming that Pre-Consultation Dialogue is a summarization task instead of a conversation QA task. Please consider changing the name to Pre-Consultation Dialogue Summarization in line 117 and other places throughout the draft, to avoid further confusion.
> >
> > Comment 2: Regarding AO5,
> > The author proposes three datasets to address the primary needs of patients at this stage. One of dataset is Pre-Consultation Dialogue (PCD), which "utilizes a multi-turn conversation to gather and summarize patients’ information" in line 095. However, PCD is only limited to evaluate the LLM summarization ability, thus the word "gather" is inappropriate.

---

> > > ### Author Rebuttal · Authors · 2024-08-21
> > >
> > > Thank you for your further questions regarding the experimental metrics and term naming. These are crucial for improving our manuscript. Below, we have provided a point-by-point response to your concerns, and we hope they can help address your concerns.
> > >
> > >
> > >
> > > **Question 1:** Regarding AO1, I find the evaluation metrics used in the study to be somewhat simplistic, especially considering that metrics are a crucial aspect of a benchmark. Do the B-4 scores or GPT-4 ratings align with human preferences? If I were to develop a language model that outperforms yours based on B-4 scores, would you or your hospital adapt to my model? Additionally, could the authors provide more details about the methodology behind the GPT-4 ratings? For example, how does GPT-4 determine which of the two outputs is superior?
> > >
> > >
> > >
> > > **Reply to Q1:**
> > >
> > >
> > > Thank you for your valuable suggestions. In response, we have carried out a human evaluation of 10 Language Learning Models (LLMs) across 7 generation tasks. For this evaluation, we instructed the annotators to assign a score from 1 to 100, based on the following criteria:
> > >
> > > ```
> > > Based on the 3 criteria below, rate the model performance on a scale of 1-100.
> > >
> > > Accuracy: The response provided by the large language model is accurate and has no factual errors. Conclusions are not made arbitrarily.
> > >
> > > Helpfulness: The model's response provides the patient with clear, instructive and practical assistance, specifically addressing the medical task.
> > >
> > > Linguistic Quality: The response logical. The model correctly understands the medical task, and the expressionis smooth and natural.
> > > ```
> > > The table displays the average score on models and task.
> > > | Model     | PCDS   | HQA   | DRG   | PDDS  | DQA   | IQA   | MQA |
> > > |-----------|-------|-------|-------|-------|-------|-------|-------|
> > > ChatGLM3|87|46|61|83|75|37|72
> > > QWen-7B|96|65|70|87|78|53|73
> > > QWen-14B|97|64|61|88|77|56|73
> > > QWen-32B|96|65|74|87|79|62|70
> > > QWen-72B|97|62|73|86|79|51|72
> > > ChatGPT|94|59|66|80|75|46|79
> > > GPT-4|98|68|76|91|78|64|82
> > > HuatuoGPT2-7B|90|56|68|76|78|49|70
> > > huatuoGPT2-34B|96|61|74|86|79|58|73
> > > DISC-MedLLM|68|42|67|80|71|39|64
> > >
> > > The rankings of 10 Language Learning Models (LLMs), based on the average metrics across the 7 tasks, are as follows:
> > >
> > > RANK|B-4|Entity|GPT Judge|GPT Judge + Entity|Human
> > > |-----------|-------|-------|-------|-------|-------|
> > > 1|ChatGPT|GPT-4|QWen-72B|GPT-4|GPT-4
> > > 2|HuatuoGPT2-34B|QWen-32B|GPT-4|QWen-32B|QWen-32B
> > > 3|QWen-72B|QWen-7B|QWen-32B|QWen-72B|HuatuoGPT2-34B
> > > 4|QWen-14B|QWen-72B|QWen-7B|QWen-7B|QWen-7B
> > > 5|ChatGLM3|QWen-14B|QWen-14B|QWen-14B|QWen-72B
> > > 6|QWen-32B|HuatuoGPT2-34B|HuatuoGPT2-34B|HuatuoGPT2-34B|QWen-14B
> > > 7|QWen-7B|ChatGPT|HuatuoGPT2-7B|ChatGPT|ChatGPT
> > > 8|GPT-4|HuatuoGPT2-7B|ChatGPT|HuatuoGPT2-7B|HuatuoGPT2-7B
> > > 9|HuatuoGPT2-7B|ChatGLM3|ChatGLM3|ChatGLM3|ChatGLM3
> > > 10|DISC-MedLLM|DISC-MedLLM|DISC-MedLLM|DISC-MedLLM|DISC-MedLLM
> > >
> > >
> > >
> > > If we compare the rankings of each pair of LLMs to see if they align with human ratings, we find that the correlation between Human and GPT Judge is 84.4%; between Human and Entity, it's 73.3%; and between Human and B-4, it's 35.5%.
> > >
> > > When we combine the scores from GPT-Judge and Entity (calculated as GPT-Judge score/100 + Entity score), the alignment improves to **91.1%**. This combined metric provides a robust measure for evaluating the performance of a Language Learning Model in the medical domain.
> > >
> > > For the details of GPT-4 as a judgement, we let it provide rating ranging from 1 to 100.
> > > The prompt used in GPT-4-as-a-judge evaluation is as follows:
> > > ```You are a professional, impartial, and strict scorer.
> > >
> > > Below is a medical task with responses from both the large language model and the ground-truth provided by human annotation.
> > > Based on the 3 criteria below, rate the model's performance on a scale of 1-100. Only provide the scores without explanations.
> > >
> > > Accuracy: The response provided by the large language model is accurate and has no factual errors. Conclusions are not made arbitrarily.
> > >
> > > Helpfulness: The model's response provides the patient with clear, instructive and practical assistance, specifically addressing the medical task.
> > >
> > > Linguistic Quality: The response logical. The model correctly understands the medical task, and the expressionis smooth and natural.
> > >
> > > Please ensure that you do not let the length of the text influence your judgment, do not have a preference for any AI assistant names that might appearin the dialogue, do not let irrelevant linguistic habits in the conversation influence your judgment, and strive to remain objective.Your scoring should be strict enough and do not give a perfectscore easily.
> > >
> > > ```

---

> > > > ### Author Rebuttal · Authors · 2024-08-21
> > > >
> > > > **Question 2:** Regarding AO2, B-4 score of about 4.0 or entity recall of about 30% are relatively low scores. This result represents that LM are not in agreement with doctor generated references. Could author provide case study to find disagreements/issues of current best model? This would pave the road for future works in this direction.
> > > >
> > > >
> > > >
> > > > **Reply to Q2:**
> > > >
> > > > Thanks for the constructive suggestion.
> > > > A common type of error in named entity recognition is mismatching.
> > > >
> > > >
> > > > Taking a case from Doctor Response Generation (DRG) as an example (translated from Chinese to English):
> > > >
> > > >
> > > > **Ground-truth:** What have you eaten in the past two days? How many times a day do you have the runs? Do you feel nauseous, vomit, or have stomach pain?
> > > > **Entities:** have the runs, Nauseous, Vomit, Stomach pain
> > > >
> > > > **Prediction:** May I ask how often you have diarrhea, and do you feel nauseous or vomit?
> > > > **Entities:** Diarrhea, Nauseous, Vomit
> > > >
> > > > In this case, "have the runs" and "diarrhea" are synonyms and should be treated as the same entity. However, they are currently mismatched.
> > > >
> > > > To address such issues, we could employ the following strategies to handle responses that don't exactly match the annotated entities but aren't necessarily incorrect:
> > > >
> > > > Post-processing and normalization: For the extracted medical entities, we can obtain synonyms for each entity based on synonym lists of disease names, symptom terms, and drug names. If any of these synonyms appear in the sentence, we consider it a successful recall.
> > > >
> > > > Vector similarity matching: We can use GPT-4 to extract medical entities from the predicted text and the BERT model as the encoder to obtain the entity embeddings (or GPT embedding). If the cosine similarity between the predicted entity and the ground-truth entity is greater than 0.8, we consider it a successful recall.
> > > >
> > > > In future work, we plan to incorporate entity normalization for a more accurate evaluation.
> > > >
> > > > **Comment 1:** Regarding ACO2, Thank the author for confirming that Pre-Consultation Dialogue is a summarization task instead of a conversation task. Please consider changing the name in line 117 and other places throughout the draft, to avoid further confusion.
> > > >
> > > > **Reply to comment 1:**
> > > >
> > > > Thanks for your helpful suggestion, we will change th name to Pre-Consultation Dialogue Summary (PCDS) to avoid the confusion in next revision.
> > > >
> > > > **Comment 2:** Regarding AO5, The author proposes three datasets to address the primary needs of patients at this stage. One of dataset is Pre-Consultation Dialogue (PCD), which "utilizes a multi-turn conversation to gather and summarize patients’ information" in line 095. However, PCD is only limited to evaluate the LLM summarization ability, thus the word "gather" is inappropriate.
> > > >
> > > >
> > > > **Reply to comment 2:**
> > > >
> > > > Thanks for your great advice, we will change from "utilizes a multi-turn conversation to gather and summarize patients’ information" to "summarize patients’ information from  a multi-turn conversation".
> > > >
> > > > We hope the above response can help address your concerns, and we would love to discuss them more in-depth if you have further questions.

---

> ### Author Response · Authors · 2024-08-27
>
> Dear Reviewer G1cc,
>
> We are genuinely grateful for your insightful comments. We have tried our best to address your initial remarks and the follow up questions, and we sincerely hope that our responses have been satisfactory.
>
> If there are any additional questions or points you wish to raise, please do not hesitate to do so. We are more than willing to engage in further discussions.
>
> Best Regards!
>
> Authors of Submission 1582

---

### Official Review · Reviewer_1xRj · 2024-07-22
**An extensive chinese clinical benchmark**

**Rating:** 7
**Confidence:** 4

**Review:**

The paper is well-written and proposed a interesting benchmark representing a wide-range of real-world tasks. Despite some minor limitations and the lack of perspective on the tasks evaluation and limitations of the datasets, the paper remain relevant and can help the community into improving LLMs understanding.

**Strengths:**

- The paper focus on a real-world pipeline and focus helpfull tasks.

- A selection of a wide range of tasks, across each of the 4 stages. Including the curation of new tasks.

- zero- vs. one-shot comparison.

- Interesting NER-based evaluation of generative outputs.

**Additional Feedback:**

N/A

**Clarity:**

The paper is clear and well written.

- "MedJourney is in Chinese" L.63: It will be kind if you say it before and in the abstract, to understand the focus of the paper.

- Some repetitions at the begining: "clinical journey into four stages. For each stage, we introduce multiple tasks." L.53 and "We divide the clinical journey into four stages." L.55

- Typo L.64 "LlamA"

- Lack of citations for some models like in the L.62 "like ChatGPT and GPT-4 but also open-source LLMs, like QWen and ChatGLM"

**Correctness:**

- "GPT-4" L.267: Why not using an already existing NER tagger for healthcare ? Or even an UMLS tagger ?

- "Since MedJourney is in Chinese, we didn’t include the English-centric LLMs, like LlamA." L.63: This is not a valuable reason since medical models like BioMistral contains chinese. The foundations models like Mistral and LLaMa also contains chinese in relatively large quantity. Having this type of models in your paper will improve our understanding of models generalization.

**Documentation:**

The authors take care to make an ethics and limitation section. But we lack of informations on the anonymisation process.

**Ethics:**

It lack of information about potential bias (genre, sex, location, age, ethnical) related to the collection process.

**Limitations:**

- "recommend the most appropriate examination" L.176: In the case of all the recommandation tasks you mentionned in this paper, you have to ensure not only if the recommendation is relevant, but also to ensure if the proposed recommendation are appropriate depending on the severity of the injuries but also on the order of delivery of medical care depending on their irreversibility or risks for the patient. For example you can cure an infection to the leg by either giving a prescription for antibiotics or by cutting off the leg (amputation). Both are valid acts, but one should be consider only as the last resort. This type of knowledge about LLMs errors is much more valuable than only considering the task as a binary task (good or wrong).

**Opportunities For Improvement:**

- "receive ethical approval from hospitals" L.59: It will interesting to have more insights of how do you manage to keep the documents compliant with ethical and privacy requirements of ACL.

- "safety" L.194 and "Understanding the indications, contraindications, side effects, dosage instructions, and potential interactions of medications is essential for safe and effective self-care." L.210: Do you consider MED-MED interaction validation in your evaluation process ?

- "GPT-4 achieves near-optimal performance on almost all tasks" L.270: What are the most frequent type of errors according to the named entities ? How do you manage answers which are not extactly matching the entities annotated but which are at the same time not wrong ?

**Relation To Prior Work:**

Well explored, excepted for non-chinese large languages models.

**Summary And Contributions:**

In this paper the authors introduce MedJourney, a new benchmark designed to evaluate the performance of large language models (LLMs) based on a patient's clinical journey in Chinese. The journey is divided into four stages, with each stage featuring multiple tasks for LLMs, accompanied by corresponding test sets. Of the 12 datasets included in MedJourney, seven are sourced from existing corpora, while five are newly created. Alongside with the benchmark, the evaluation of multiple open-source, closed-source and chinese medical LLMs was performed on each of the tasks in zero- and one-shot scenarios.

---

> ### Author Rebuttal · Authors · 2024-08-17
>
> Thanks for your helpful comments, below are the replies to your questions. In the following O denotes oppotunity, L denotes Limitation,  CO denotes Correctness and CL denotes Clarity. And AO, AL, ACO, ACL denote the answers to O, L, CO and CL respectively.
>
> **O1**: More insights of how do you manage to keep the documents compliant with ethical and privacy requirements
>
> **AO1**: In this paper, we present 12 datasets, seven of which are constructed from existing public datasets.
>
> Regarding the newly proposed datasets, Hospital Reception QA (HQA), Drug QA (DQA), and Insurance QA (IQA) primarily focus on medical knowledge and do not contain patients' personal information. For instance, as demonstrated in Table 5 in the paper, and Tables 2 and 7 in the supplementary documents, the questions pertain to drug usage, hospital visitation knowledge, and insurance. These datasets contain minimal personal information, a fact we have confirmed with human experts.
>
> The Department Recommendation (DR) and Pre-Consultation Dialogue (PCD) datasets are derived from users' inquiries prior to doctor visits. Therefore, these datasets focus on patients' primary complaints and do not include examination, diagnosis, or medication information.
>
> For the Department Recommendation (DR) and Pre-Consultation Dialogue (PCD) datasets, we also employ experienced physicians to recompose the data instead of directly using patients' input, further ensuring the absence of patient information.
>
>
> For example,
>
> |orignial patients' data	|rewritten patients' data|
> |-----------|-------|
> |My complexion is yellowish, and I have low physical strength. I easily get short of breath and frequently feel nauseous.|	My skin appears yellowish, and I experience an accelerated heartbeat and shortness of breath even with slight exertion. I constantly feel a bit nauseous. Which medical specialty should I consult?|
> |My mother is 70 years old. She started experiencing sudden pain in her knees 10 days ago, and it has persisted without improvement even after resting for several days. In fact, it has worsened. Which medical specialty should she consult?	|Elderly person with sudden knee joint pain and limited mobility. Symptoms have worsened over the past 10 days.|
>
> Even though we did not directly use patients' data, we still obtained ethical approval from the hospitals.
>
>
>
>
> **O2**: Do you consider MED-MED interaction validation in your evaluation process
>
> **AO2**: The distribution of intent categories in **DQA** dataset are as follows:
> - Indications: 27.0%
> - Adverse Reactions: 21.9%
> - Dosage Instructions: 18.2%
> - Traditional Chinese Medicine: 10.2%
> - Special Populations: 8.0%
> - Precautions: 6.5%
> - Pharmacokinetics: 3.6%
> - Contraindications: 3.0%
> - Interactions: 1.6%
>
> Therefore the MED-MED interaction is included.
>
> **O3**: What are the most frequent type of errors according to the named entities?
>
> **AO3**：The most frequent type of error according to the named entities is the omission of named entities. Taking a case from Doctor Response Generation (DRG) as an example (translated to English, may not be accurate):
>
> Ground-truth: What have you eaten in the past two days? How many times a day do you have the runs? Do you feel nauseous, vomit, or have stomach pain?
>
> Entities: have the runs, Nausea, Vomiting, Stomach pain
>
> Prediction: May I ask how often you have diarrhea, and do you feel nauseous or vomit?
>
> Entities: Diarrhea, Nausea, Vomiting
>
> Since “have the runs” and “diarrhea” are synonyms, they should be treated as the same words. We will add the entity normalization in future work for a more accurate evaluation.
>
>
>
>
>
> **L1:** recommend the most appropriate examination
>
> **AL1:**
> Thank you for your valuable suggestions. The tasks of Examination Prediction (EP), Disease Prediction (DP), Treatment Prediction (TP), and Medication Prediction (MP) are designed to recommend potential actions for doctors, such as suggesting further examinations, possible diagnoses, appropriate treatments, and medications for patients. These four tasks are derived from the China National Medical Licensing Examination (CNMLE), a crucial qualifying examination for becoming a certified doctor in China. The CNMLE covers knowledge from 20 medical subjects across four areas: clinical medicine, preclinical medicine, medical humanities, and preventive medicine. Therefore, in the CNMLE, cases are also considered that select the most suitable treatment according to the severity of patients' symptoms. For example,
>
> > Question: A 70-year-old male has been experiencing difficulty swallowing for three months. An esophagram revealed a 3cm irregular filling defect in the middle segment of the esophagus, and a biopsy of the tissue confirmed squamous cell carcinoma. His electrocardiogram showed ST-segment depression and T-wave inversion. Pulmonary function tests indicated severe impairment. What is the appropriate treatment method? Possible treatment options include: drug therapy, radical surgery, palliative surgery, radiation therapy, and gastrostomy. Please choose the most correct treatment from the above options.
>
> >Answer: Radiation therapy
>
> The patient is of advanced age (70 years old), and his electrocardiogram indicates myocardial ischemia (ST-segment depression and T-wave inversion). His pulmonary function is severely impaired. These conditions make the patient unsuitable for surgical treatment (whether radical or palliative) due to the high risk. At the same time, the choice of drug therapy needs to be very cautious for elderly patients, especially those with poor renal function. In this situation, radiation therapy becomes a relatively safe and effective treatment method.
>
> In summary, Treatment Prediction (TP) considers patients' condition to select the most proper treatment.

---

> > ### Author Rebuttal · Authors · 2024-08-17
> >
> > **CO1:** Why not using an already existing NER tagger for healthcare.
> >
> > **ACO1:** We utilize entity extraction to evaluate the performance of generation tasks in addition to the NLG metrics. This requires extracting medical entities from patient-doctor dialogues (DRG), medical editors (DQA), and insurance specialists (IQA). To handle texts from different sources, we employ GPT-4, which is relatively robust across various text forms. Additionally, since the text to be processed is in Chinese, we cannot directly use a UMLS tagger.
> >
> >
> > **CO2:** Evalute on more LLMs.
> >
> > ACO2: Thank you for your suggestion. we have supplemented our results with Alpaca2, an enhanced version of LlamA2 optimized for Chinese vocabulary. We have also introduced Mistral NeMo 12B as a multilingual LLM baseline. The results are reported in the following table.
> >
> > We also attempt to conduct experiments on LlamA2. However, when we input a Chinese text for a disease prediction task, LlamA2 returned a translated English version without providing an answer to the question. Therefore, we experimented with Alpaca2 instead.
> >
> >
> > | Model | DR Acc| PCD B-4|HQA B-4|PDDS B-4|DRG B-4|	EP Acc|	DP Acc|	TP Acc|	MP Acc|	DQA B-4|IQA B-4|MQA B-4|AVG Acc|AVG B-4|
> > |:------:|:------:|:------:|:------:|:------:|:-----:|:-----:|:-----:|:-----:|:-----:|:-----:|:-----:|:-----:|:-----:|:-----:|
> > | Alpaca2 7B (zero-shot)| 	0.168| 	6.623| 	2.392| 	14.054| 	2.33	| 0.248| 	0.239| 	0.135| 	0.232| 	5.226| 	4.814| 	3.362| 0.2044	|5.543|
> > Alpaca2 7B (one-shot)| 	0.176| 	8.755| 	4.179| 	20.312| 	2.723| 	0.201	| 0.202| 	0.068| 	0.174| 	5.431| 	4.959| 	3.046| 0.1642	|7.057857143|
> > Mistral NeMo 12B (zero-shot)	| 0.236| 	6.84| 	2.608| 	7.922| 	2.536	| 0.41| 	0.35| 	0.365| 	0.313| 	6.21| 	4.315| 	3.592|0.3348|	4.860428571|
> > Mistral NeMo 12B (one-shot)	| 0.36| 	23.378| 	4.609| 	13.697| 	2.916	| 0.4	| 0.386| 	0.277	| 0.362| 	5.975| 	4.638| 	3.798| 0.357|	8.430142857|
> >
> >
> >
> > **CL1:** Typo and missing reference.
> > **ACL1:** Thanks, we will fix the typos and add references.

---

### Official Review · Reviewer_rzWr · 2024-07-23
**MedJourney review**

**Rating:** 6
**Confidence:** 4

**Review:**

This study introduces the first comprehensive benchmark for evaluating LLMs across the entire patient clinical journey, making a significant contribution to the field. Its strengths lie in the diverse, physician-annotated dataset covering 12 tasks across four care phases, providing a clinically relevant evaluation framework. However, the research has notable limitations. These include a lack of transparency in scenario selection criteria and insufficient representation of longitudinal care and stage interconnectedness. Due to these shortcomings, the study's claims of real-world applicability are difficult to accept fully.

- Pros:
1. Comprehensive patient journey coverage
2. Diverse, clinically relevant dataset
3. Multi-stage evaluation framework

- Cons:
1. Lack of transparent selection criteria
2. Insufficient longitudinal and interconnected assessment
3. Questionable real-world applicability claims

**Strengths:**

This study makes significant contributions to the medical AI field. It aims to present a comprehensive benchmark covering the entire patient clinical journey, offering a structured framework for assessing LLMs in healthcare contexts. The inclusion of 4,960 cases across 12 diverse tasks, spanning 4 distinct care phases, provides a broad evaluation landscape. The involvement of experienced physicians in dataset annotation enhances the clinical relevance and quality of the benchmark.

**Additional Feedback:**

n/a

**Clarity:**

While the overall structure of the paper is sound, the lack of specific details about the dataset's composition and capabilities significantly hampers its clarity. More explicit information about the medical conditions covered, the types of evaluations possible, and concrete examples of scenarios would greatly enhance the paper's clarity and utility for the research community.

**Correctness:**

While the study presents a novel approach to evaluating LLMs across the patient clinical journey, there are several concerns regarding the correctness of its claims and methodology:
1. Dataset Construction: The lack of transparent criteria for disease selection and scenario construction raises questions about the soundness of the dataset. Without clear guidelines, it's difficult to assess whether the benchmark truly represents the diversity and complexity of real-world medical situations, particularly for rare diseases or complex cases.
2. Evaluation Methods: The study's evaluation metrics primarily focus on text generation accuracy and similarity, which may not adequately capture critical aspects of clinical decision-making. This narrow focus potentially overlooks important elements of medical practice, such as diagnostic accuracy and treatment appropriateness.
3. Experiment Design: The study attempts to evaluate the patient's clinical journey in four stages, but fails to adequately represent the continuity and interconnectedness of these stages. This design flaw limits the assessment of LLMs' ability to maintain context and make consistent decisions across a patient's entire journey, which is crucial in real clinical settings.
4. Claims of Real-World Applicability: The study's claims of reflecting real-world clinical practice are difficult to fully accept due to the aforementioned limitations. The proposed scenarios may not accurately mirror the complexity and uncertainty of actual clinical environments, potentially overestimating the benchmark's relevance to real-world medical decision-making processes.
5. Validation: There is insufficient in-depth analysis to effectively validate the proposed datasets, particularly in demonstrating how well they represent the claimed "entire hospital visit journey." This lack of rigorous validation weakens the study's claims of comprehensive coverage of the patient clinical journey.

**Documentation:**

While the study provides some information about its dataset, there are several areas where the documentation falls short of supporting full reproducibility and understanding:

1. Data Collection and Organization:
  - The study lacks clear criteria for disease selection and scenario construction, making it difficult to understand how the dataset was composed.
  - There is insufficient detail on how the 12 tasks were developed and how they relate to real-world clinical practices.
  - The process of integrating existing datasets with newly created ones is not thoroughly explained.

2. Dataset Composition Transparency:
  - The rationale behind the selection of specific medical conditions, especially rare diseases or complex cases, is not clearly presented.
  - There's a lack of information on the distribution of different types of medical scenarios across the four stages of the patient journey.

3. Real-World Applicability:
  - There's insufficient documentation on how closely the dataset mirrors actual clinical environments, particularly in terms of complexity and uncertainty.
  - The study doesn't clearly demonstrate how the scenarios in the dataset relate to real-world medical decision-making processes.

**Limitations:**

1. Lack of Transparent Criteria and Representativeness: The study lacks clear criteria for dataset composition, particularly in disease selection and scenario construction. This ambiguity hinders a comprehensive assessment of the benchmark's representativeness, especially regarding rare diseases or complex medical situations.
2. Insufficient Longitudinal and Real-World Representation: While the study attempts to evaluate the patient's clinical journey in four stages, it fails to adequately represent the continuity and interconnectedness of these stages. This limitation, combined with the study's shortfall in mirroring the complexity and uncertainty of actual clinical environments, impedes the assessment of LLMs' ability to handle real-world medical decision-making processes and long-term patient information effectively.
3. Experimental and Validation Limitations: The evaluation metrics primarily focus on text generation, overlooking critical elements of clinical accuracy and decision-making appropriateness. Furthermore, there's insufficient in-depth analysis to validate the datasets' representation of the entire patient journey. These limitations weaken the study's claims of comprehensive clinical journey coverage and limit the practical applicability of the results in real clinical settings.

**Opportunities For Improvement:**

1. Dataset:
- Clearly define and publish the criteria used for disease selection, scenario construction, and dataset composition.
- Transparently disclose what specific aspects of medical knowledge and decision-making the benchmark is designed to evaluate. For instance, clarify if it assesses diagnostic reasoning, treatment planning, or patient communication skills.
- Publish a comprehensive guide on what researchers can expect to learn about LLM performance when using this benchmark, including its strengths and limitations in evaluating different aspects of medical AI capabilities.
- Demonstrate how the benchmark reflects the continuity of care and information flow that occurs in real clinical settings.

2. Experiments:
- Conduct an analysis to quantify how well the benchmark mirrors the complexity and uncertainty of real clinical environments. This could include comparing the distribution of case complexity in the benchmark with that seen in actual medical practice.
- Develop a scoring system that evaluates LLM performance across the entire patient journey, assessing consistency and appropriateness of decisions.

**Relation To Prior Work:**

1. Comprehensive Patient Journey Coverage: Unlike previous benchmarks that often focus on specific medical tasks or exam-style questions, this study introduces the first comprehensive benchmark covering the entire patient clinical journey. It divides the journey into four distinct stages (planning, access, delivery, and ongoing care), providing a more holistic evaluation framework for LLMs in healthcare settings.
2. Diverse and Clinically Relevant Dataset: The study presents an extensive dataset of 4,960 instances across 12 tasks, which is more diverse and comprehensive than many existing medical AI benchmarks. The inclusion of both new and existing data, annotated by experienced physicians, enhances the clinical relevance and robustness of the benchmark.
3. Multi-Stage Evaluation Approach: By assessing LLM performance across four distinct care phases, the study offers a more nuanced and realistic evaluation compared to previous single-task or single-stage assessments.

**Summary And Contributions:**

This study introduces MedJourney, the first comprehensive benchmark for evaluating LLMs across the entire patient clinical journey. The researchers divided the clinical journey into four stages: planning, access, delivery, and ongoing care, encompassing 12 diverse tasks.
The benchmark comprises 12 datasets totaling 4,960 medical cases, dialogues, and question-answer pairs. Five datasets are newly constructed and annotated by experienced physicians, while seven are reconstructed from existing sources. The datasets are distributed across four stages: 3 for planning (733 cases), 2 for access (400 cases), 4 for delivery (3,370 cases), and 3 for ongoing care (457 cases).
Various LLMs were evaluated using zero-shot and one-shot approaches, with performance assessed through accuracy, NLG metrics, and entity-level recall.

Key contributions include: (1) the first benchmark covering the patient’s clinical journey, (2) an extensive dataset of 12 tasks across four clinical stages, and (3) a comparative analysis of different LLM types in medical applications.

---

> ### Author Rebuttal · Authors · 2024-08-17
>
> Thanks for your helpful comments, below are the replies to your questions.
>
> **Q1:** Dataset: Dataset Collection and Organization. Composition Transparency:
>
> **A1:** In this paper, we segment the clinical process based on the patient's journey, specifically focusing on the stages before visiting the hospital (planning), during the hospital visit (access and delivery), and after the hospital visit (ongoing care).
>
> During the planning stage, patients experiencing symptoms are guided to the appropriate department through Department Recommendation (DR). The Pre-Consultation Dialogue (PCD) collects the patient's main complaints in a dialog format and summarizes them for the doctor's review. The Hospital Reception QA (HQA) informs patients about the specifics of their hospital visit, such as what items to bring, dietary restrictions before certain examinations, and so on.
>
> For the **DR** dataset, we have chosen 100 fine-grained departments from 12 coarse-grained departments, which include 59 adult departments and 41 pediatric departments. Our selections were informed by the Directory of Medical Institutions’ Clinical Departments and real-world department structures. For each fine-grained department, five patient complaints were manually generated, resulting in a total of 500 data entries. The distribution of 12 coarse-grained departments can be found in Figure 1 in Appendix A.1.1.
>
> For the **PCD** dataset, in order to construct a representative dataset, we have collected 77 different symptoms from 36 fine-grained departments based on real user consultations. The representative departments and symptoms include:
> - Oral Department (19%, including oral restoration, orthodontics, etc.): loose teeth, oral ulcers, impacted teeth, etc.,
> - Dermatology Department (13%): skin itching, rashes, pigmentation, etc.,
> - Pediatrics Department (12%): abnormal height, early loss of deciduous teeth, spitting up milk, cough, etc.,
> - Ophthalmology Department (8%): decreased vision, trichiasis, ptosis, etc.,
> - Gynecology Department (7%): infertility, irregular menstruation, abnormal vaginal discharge, etc.
> - Respiratory Medicine Department (5%): asthma, dyspnea, upper respiratory tract infection, etc.
>
> During the delivery stage, Examination Prediction (EP), Disease Prediction (DP), Treatment Prediction (TP), and Medication Prediction (MP) recommend potential actions for the doctor's reference, such as suggesting further examinations, possible diagnoses, appropriate treatments, and medications for the patient. These four tasks are collected from China National Medical Licensing Examination (CNMLE) which is an essential qualifying examination to become a certified doctor in China, covering knowledge from 20 medical subjects in four areas: clinical medicine, preclinical medicine, medical humanities, and preventive medicine. Therefore the rare diseases or complex cases are covered in these four datasets.
>
> In the **DP** dataset, it includes a total of 1761 question-answer pairs that cover 1490 distinct diseases. Among these, we found 150 diseases (approximately 10%) that have relatively low incidence rates globally or are relatively rare in specific populations.
>
> In the ongoing care stage, Drug QA (DQA) instructs patients on medication usage and related knowledge. Insurance QA (IQA) provides information on medical insurance, and Mental Health QA (MQA) focuses on the patient's mental well-being.
>
> For the **DQA** dataset, the questions in our dataset were extracted from real patient inquiries and subsequently filtered by experienced clinicians.
>
> The final dataset comprises 137 question-answer pairs, with the distribution of intent categories as follows:
> - Indications: 27.0%
> - Adverse Reactions: 21.9%
> - Dosage Instructions: 18.2%
> - Traditional Chinese Medicine: 10.2%
> - Special Populations: 8.0%
> - Precautions: 6.5%
> - Pharmacokinetics: 3.6%
> - Contraindications: 3.0%
> - Interactions: 1.6%
>
>
> While the proposed 12 tasks do not cover all aspects of a clinical process, such as radiology report generation, they effectively connect the dots of a patient's journey.
>
>
> **Q2:** Evaluation Methods
>
> **A2:** In addition to the NLG metrics like BLEU and the accuracy metrics, we also conduct entity based metrics in Table 8 which can evaluate whether the core information is obtained.
>
>
> **Q3:** Experiment Design
>
> **A3:** Thank you for your insightful suggestions regarding the evaluation process. The concept of continuity and interconnectedness across the four stages provides a valuable perspective for designing experiments. For instance, if we can track the same patient throughout their entire journey, we can design experiments that emphasize continuity and interconnectedness. However, in real-world clinical settings, it's rare for a patient to undergo all 12 tasks. Even for those who do, data privacy concerns and the disconnect between different systems can make it challenging to link patients across different stages.
>
> **Q4&Q5:** Claims of Real-World Applicability & Validation
>
> **A4&A5:** Many proposed tasks have been already deployed in clinics. In the Tencent Smart Hospital Solution (https://healthcare.tencent.com/en), the Department Recommendation (DR), Hospital Reception QA (HQA) and Pre-Consultation Dialogue (PCD) have been deployed to more than 1,500 hospitals in China (more than 600 are top-tier hospitals), with more than 1.5 million month active users. These three tasks are relating to patients information understanding and gathering, before visiting the doctors.
>
> we will take the LLM enabled Pre-Consultation Dialogue (PCD) deployed in Shenzhen Renmin Hospital for example, till now, it has served 100 thousands patients. On the doctor side, the PCD achieves a 75% reduction (p-value<0.001) in the total time of writting notes for doctors. As a results, the doctors have more time to meet more patients. On the patient sides, it achieves an increase of 28% in patients' saficatory degeee.

---

### Official Review · Reviewer_nvGG · 2024-07-25
**MedJourney offers an innovative and practical benchmark for evaluating LLMs in healthcare, with a detailed and ethically grounded approach but limited by its focus on Chinese datasets.**

**Rating:** 7
**Confidence:** 4
**Correctness:** Yes
**Clarity:** Yes

**Review:**

The paper is well-structured, clearly outlining the MedJourney benchmark, which aligns tasks and datasets with stages of a patient’s clinical journey. The evaluation methodology for various LLMs is detailed and clear. The work is original in its approach to align benchmarks with practical aspects of patient care, making it highly relevant for healthcare applications. This study is significant for its potential to enhance AI applications in healthcare by providing a relevant and structured testing environment.

Pros

	•	Covers a wide range of real-world patient interactions.
	•	Thoroughly assesses both general and specialized LLMs.
	•	Rigorously addresses ethical aspects of AI in healthcare.

Cons

	•	Focuses primarily on Chinese LLMs, limiting wider applicability.
	•	May be challenging to replicate in smaller or less resourced settings.
	•	Requires substantial computational resources for full evaluation.

**Strengths:**

See Review section

**Additional Feedback:**

N/A

**Documentation:**

Yes

**Opportunities For Improvement:**

N/A

**Relation To Prior Work:**

Yes

**Summary And Contributions:**

The paper introduces “MedJourney,” a benchmark for evaluating Large Language Models (LLMs) throughout a patient’s clinical journey in healthcare settings. It segments the patient journey into four stages—planning, access, delivery, and ongoing care—and provides 12 datasets for assessing LLMs across these stages. The benchmark includes both existing and newly created datasets, offering a comprehensive tool to test LLMs’ effectiveness in real-world medical scenarios. Key contributions are the structured evaluation of both general and specialized medical LLMs and the ethical and professional validation of the datasets used.

---

> ### Author Rebuttal · Authors · 2024-08-17
>
> Thanks for your helpful comments, below are the replies to your questions.
>
>
> **Q1:** Focuses primarily on Chinese LLMs, limiting wider applicability.
>
>
> **A1:** Thank you for your insightful suggestion. In our manuscript, we evaluated several public large language models (LLMs) such as ChatGLM3 and QWen, as well as specialized LLMs like HuatuoGPT2 and DISC-MedLLM, which are primarily Chinese LLMs. We fully acknowledge your point about the broader applicability of our benchmark. To address this, we conducted additional experiments using Alpaca2 (an enhanced version of Llama2 optimized for Chinese vocabulary) and MISTRAL. The specific results are as follows:
>
> | Model | DR Acc| PCD B-4|HQA B-4|PDDS B-4|DRG B-4|	EP Acc|	DP Acc|	TP Acc|	MP Acc|	DQA B-4|IQA B-4|MQA B-4|AVG Acc|AVG B-4|
> |:------:|:------:|:------:|:------:|:------:|:-----:|:-----:|:-----:|:-----:|:-----:|:-----:|:-----:|:-----:|:-----:|:-----:|
> | Alpaca2 7B (zero-shot)| 	0.168| 	6.623| 	2.392| 	14.054| 	2.33	| 0.248| 	0.239| 	0.135| 	0.232| 	5.226| 	4.814| 	3.362| 0.2044	|5.543|
> Alpaca2 7B (one-shot)| 	0.176| 	8.755| 	4.179| 	20.312| 	2.723| 	0.201	| 0.202| 	0.068| 	0.174| 	5.431| 	4.959| 	3.046| 0.1642	|7.058|
> Mistral NeMo 12B (zero-shot)	| 0.236| 	6.84| 	2.608| 	7.922| 	2.536	| 0.41| 	0.35| 	0.365| 	0.313| 	6.21| 	4.315| 	3.592|0.3348|	4.860|
> Mistral NeMo 12B (one-shot)	| 0.36| 	23.378| 	4.609| 	13.697| 	2.916	| 0.4	| 0.386| 	0.277	| 0.362| 	5.975| 	4.638| 	3.798| 0.357|	8.430|
>
>
> We also attempt to conduct experiments on LlamA2. However, when we input a Chinese text for a disease prediction task, LlamA2 returned a translated English version without providing an answer to the question. Therefore, we experimented with Alpaca2, which is a Chinese-enhanced version of LlamA2.
>
>
>
>
>
>
> **Q2 & Q3:** May be challenging to replicate in smaller or less resourced settings.  Requires substantial computational resources for full evaluation.
>
> **A2 & A3:** The proposed MedJourney is designed to evaluate the performance of Large Language Models (LLMs) in the medical domain. The computational resources required depend on the specific model being evaluated.
>
> To evaluate Alpaca2-7B using a single V100, it takes a total of 198 minutes to complete all twelve tasks in both zero-shot and one-shot settings.
>
> To evaluate Mistral NeMo 12B using two V100s, it takes a total of 1034 minutes to complete all twelve tasks in both zero-shot and one-shot settings.
>
> To evaluate a 72b model using eight V100 GPUs, it takes less than a day to complete all twelve tasks in both zero-shot and one-shot settings.
>
> The average time required for each question in each task is presented below (in seconds).
> | Model | DR| PCD|HQA|PDDS|DRG|	EP|	DP |	TP|	MP|	DQA|IQA|MQA|
> |:------:|:------:|:------:|:------:|:------:|:-----:|:-----:|:-----:|:-----:|:-----:|:-----:|:-----:|:-----:|
> |Alpaca2-7b|    0.24|	4.85	|3.91	|5.65	|1.29|	0.36	|0.32|	0.84|	0.32|	3.08|	2.32|	9.09|
> |MistralNeMo-12B|	1.10|	18.70|	14.80	|33.00|	5.00|	0.90	|1.30	|1.30|	1.00	|31.70	|28.80|	38.90|
> |Qwen-72b|	2.97|	63.23|	41.98|	68.27|	25.04|	3.18|	4	|3.75	|3.14|	52.04|	51.13|	86.02|

---

### Decision · Program_Chairs · 2024-09-26

**Decision:**

Accept (Poster)

**Comment:**

This work proposes a dataset that aims to evaluate an LLM’s ability in addressing diverse phases of patient treatment in a clinical environment. The dataset consists of 12 subtasks that covers the entire “patient journey”, such as department recommendation, pre-consultation, reception, doctor-patient dialogue, etc. The dataset was constructed based on real-world patient records collected from collaborating hospitals, and clinicians were involved in processing the final dataset. Although the reviewers had concerns regarding insufficient description of selection criteria and clinical process, and the original dataset being Chinese, the fact that the authors focused on evaluating LLMs with respect to multiple stages of patient treatment in hospitals, and constructed a high-quality evaluation dataset using real-world medical records is admirable, and the dataset is likely to have a positive impact in the AI community.